# Neural Network Differential Equation Solvers allow unsupervised error analysis and correction

## Abstract

Neural Network Differential Equation (NN DE) solvers have surged in popularity due to a combination of factors: computational advances making their optimization more tractable, their capacity to handle high dimensional problems, easy interpretability, etc. However, most NN DE solvers suffer from a fundamental limitation: their loss functions are not explicitly dependent on the errors associated with the solution estimates. As such, validation and error estimation usually requires knowledge of the true solution. Indeed, when the true solution is unknown, we are often reduced to simply hoping that a "*low enough*" loss implies "*small enough*" errors, since explicit relationships between the two are not available. In this work, we describe a general strategy for efficiently constructing error estimates and corrections for Neural Network Differential Equation solvers. Our methods do not require *a priori* knowledge of the true solutions and obtain explicit relationships between loss functions and the errors. In turn, these explicit relationships allow for the unsupervised estimation and correction of the model errors.

## 1 Introduction

Deep learning has heralded new methods for many scientific disciplines – the field of numerical methods for differential equations has been no exception (1; 2; 3; 4). Deep neural network based differential equations (NN DE) solvers have been proposed under a variety of different names (PINNs (2), DGM (3), etc) – each catering to various classes of problems. However, they all share certain common features: the use of a known differential equation (DE) as the central component of an appropriate loss function, the use of existing knowledge (boundary conditions, experimental/synthetic data, etc) to constrain the search for solutions, randomized optimization methods that sample from the domain of interest at a requisite resolution, etc.

We investigate another common facet of many NN DE solvers: the lack of unsupervised error quantification/correction methods to estimate model errors without prior knowledge of the solution. Most solvers use the equation based loss functions as a surrogate measure for the error. However, while such a measure is intuitively related to the error in the solution model, an explicit description of that connection is mandatory, if error quantification is to be done without knowing the solution.

We achieve these goals by explicitly relating the loss terms and the model error. We showcase how these connections allow results on the model error that don't rely on prior knowledge of solution. We propose techniques by which these results can be used to build significantly more efficient NN DE solvers, with only marginal increases in computational complexity. We formalize our ideas into four theorems, two inequalities, and two algorithms. We validate our claims with a collection of numerical experiments on several non-trivial DEs (including nonlinear PDEs).

For the sake of readability and simplicity, all proofs have been rigorously presented in the appendices, while the main text simply reports the results and discusses their significance.

An associated codebase is also provided, with inbuilt options for the DEs already studied as part of this work. However, the codebase has been designed so that the users may easily add their own DEs of interest (the assumptions under which this work is valid should be general enough for a wide variety of DEs from many different scientific disciplines).

## 2 NEURAL NETWORK DIFFERENTIAL EQUATION SOLVERS

For ease of discussion, we let $\mathcal{D} \subseteq \mathbb{R}^d$ denote some closed, bounded, path connected domain of interest for some differential equation (DE). Let $\partial\mathcal{D} \subseteq \mathcal{D}$ be the portion of the domain over which some constraint conditions on the solution exist (usually obtained as boundary conditions, empirical data, etc). Assume that our chosen DE, when given unique constraints over $\partial\mathcal{D}$, admits a unique solution $\Phi : \mathcal{D} \to \mathbb{R}^D$.

We wish to consider a (possibly non-linear) equation operator $\mathbf{F} : G \to H$, where $G$ and $H$ are some suitable spaces of functions over $\mathcal{D}$ ($\Phi \in G$). We decompose $\mathbf{F}$ as

$$\mathbf{F}[\bullet] = \mathbf{L}[\bullet] + \mathbf{N}[\bullet] + \mathbf{C} \tag{1}$$

where $\mathbf{L}$ represents the term(s) which depend linearly on $\Phi$, $\mathbf{N}$ the represents the term(s) which depend non-linearly on $\Phi$, and $\mathbf{C}$ are the terms independent of $\Phi$. This additive decomposition into linear, nonlinear, and constant terms is always possible: for linear DEs, $\mathbf{N} \equiv 0 \equiv \mathbf{C}$. We have

$$\mathbf{F}[\Phi] = \mathbf{L}[\Phi] + \mathbf{N}[\Phi] + \mathbf{C} = 0 \ .$$

Let us assume we wish to construct an NN based approximation $\mathcal{N} : \mathcal{D} \to \mathbb{R}^D$. Let us also assume the NN uses analytic activation functions so that $\mathcal{N} \in \mathcal{C}^\infty$. Let $W$ be its width (neurons per hidden layer) and $D_\mathcal{N}$ be its depth (number of hidden layers). Finally, let $\mathbf{w} \equiv \{b_1^1, w_{11}^1, w_{12}^1, \ldots, b_1^2, w_{11}^2, w_{12}^2, \ldots\} \in \mathcal{M} \subseteq \mathbb{R}^M$ be the $M$ weights and biases of the NN.

### 2.1 EXISTENCE AND COMPLEXITY

Feedforward NNs with ReLU activation and $W \geq d + 4$ can arbitrarily well approximate any Lebesgue integrable function $\Phi : \mathcal{D} \to \mathbb{R}^D$ w.r.t. $L^1$ norm, provided that $\mathcal{D}$ is some compact subset of $\mathbb{R}^d$ and enough depth $D_\mathcal{N}$ is provided to the NN (5). The same holds for NNs with ReLU activation and $D_\mathcal{N} \geq \log_2(d + 1)$ provided that they are wide enough (6). Hence, there exist $\mathbf{w}$ such that $\mathcal{N}$ can arbitrarily approximate any $\Phi \in \mathcal{C}^k$ over $\mathcal{D}$, given large enough $(W, D_\mathcal{N})$.

These theoretical guarantees may be practically realized by leveraging the minimum regularity expected from $\Phi$. Recall $\mathcal{W}^{k,\infty}(\mathcal{D})$ is the Sobolev space of order $k$, on compact $\mathcal{D} \subseteq \mathbb{R}^d$, w.r.t. the $L^\infty$ norm. Let us assume that $\Phi \in \mathcal{W}^{k,\infty}(\mathcal{D})$. There exists some $\mathcal{N}$ with $(W, D_\mathcal{N})$ that can $\varepsilon$ - approximate $\Phi : \mathcal{D} \to \mathbb{R}$, if $W = d + 1$, $D_\mathcal{N} = \mathcal{O}(\mathrm{diam}(\mathcal{D})/\omega_f^{-1}(\varepsilon))^d$ (7), where

$$\omega_f^{-1}(\varepsilon) = \sup\{\delta : \omega_f(\delta) \leq \varepsilon\}, \quad \delta = |\mathbf{x}_1 - \mathbf{x}_2|, \quad \mathbf{x}_1, \mathbf{x}_2 \in \mathcal{D}.$$

There also exist $\mathcal{N}$ that $\varepsilon$ - approximate $\Phi : \mathcal{D} \to \mathbb{R}$, if $M = \mathcal{O}(ln(\frac{1}{\varepsilon})\varepsilon^{-d/k})$, $D_\mathcal{N} = \mathcal{O}(ln(\frac{1}{\varepsilon}))$ (8).

### 2.2 OPTIMIZATION

The loss function $\mathcal{L}$ to train $\mathcal{N}$ usually takes the following form (2; 3; 4):

$$\mathcal{L} = \mathbb{E}_{x_1 \in \mathcal{D}}\left[\|\mathbf{F}[\mathcal{N}(x_1)]\|_p\right] + \mathbb{E}_{x_2 \in \partial\mathcal{D}}\left[\|\Phi(x_2) - \mathcal{N}(x_2)\|_p\right] \tag{2}$$

where $\|\cdot\|_p$ is the usual $p$-norm on $\mathbb{R}^D$. Variants of Stochastic Gradient Descent (SGD) are *almost always* capable of *eventually* reaching adequately small loss values for large enough $(W, D_\mathcal{N})$ under such loss (9). Thus as $\mathcal{L} \to 0$ during optimization, the uniqueness of $\Phi$ implies $\mathcal{N} \to \Phi$ over $\mathcal{D}$ (formally shown in Theorem 1).

### 2.3 ALTERNATIVE PARAMETERIZATION OF THE CONSTRAINT CONDITIONS ON $\partial\mathcal{D}$

Sometimes, the constraint conditions (such as initial or boundary conditions) may be enforced by using the following parametrization as the model $\mathcal{N}$ for $\Phi$, at some arbitrary $x_1 \in \mathcal{D}$:

$$\mathcal{N}(x_1) = \Phi(x_2) + \mathrm{dist}(x_1, x_2)\mathcal{N}^o \tag{3}$$

where $x_2 \in \partial\mathcal{D}$ is some appropriate *nearest* constraint point to $x_1$. Here, $\mathcal{N}^o$ represents the NN and $\mathrm{dist}(x_1, x_2)$ is some metric that enforces that the model $\mathcal{N}$ is always exact over $\partial\mathcal{D}$, while allowing the NN flexibility to *learn* $\Phi$ elsewhere (ex: $1 - e^{-\|x_1 - x_2\|}$, as used in (10)). This parametrization can eliminate the need for the $\mathbb{E}_{x_2 \in \partial\mathcal{D}}$ term in Eq. 2. Our work is applicable in such scenarios too.

## 3   ERROR BOUNDS AND CONVERGENCE FOR NN DE SOLVERS

Define $\Phi_{\epsilon_1} := \Phi - \mathcal{N}$ as the error in our solution model. Three simple observations about $\mathcal{N}, \Phi_{\epsilon_1}$, the convergence of $\mathcal{L}$ to 0, and their inter-plays underpin the entirety of this work:

1. $\mathbf{F}[\mathcal{N}]$ is not explicitly dependent on $\Phi_{\epsilon_1}$: there is no explicit relation between $\mathcal{L}$ and $\Phi_{\epsilon_1}$ over $\mathcal{D} - \partial\mathcal{D}$, where $\Phi$ is unknown. We don't know **if/how** $\mathbb{E}\left[\|\Phi_{\epsilon_1}\|\right] \to 0$ as $\mathcal{L} \to 0$.

2. Thus, $\Phi_{\epsilon_1}$ associated with $\mathcal{N}$ is not estimable over $\mathcal{D} - \partial\mathcal{D}$ in standard NN DE solvers.

3. Optimization performance saturates when $\mathcal{N}$ settles around a local minima(s) of $\mathcal{L}$.

In its most general form, the first problem can be immensely intractable and is often handled separately for different kinds of DEs (3; 10; 11; 12; 13). In this section, we present generic results (proofs in Appendix B) which strongly combat this problem for a large collection of DEs originating in scientific domains, including many nonlinear PDEs. Our results will be valid for any $\mathbf{F}[\mathcal{N}]$, s.t. $\mathbf{F}[\mathcal{N}] = 0 \implies \mathcal{N} = \Phi$, thus allowing a larger class of loss function designs as well.

**Theorem 1.** *Let $G$ and $H$ be two Banach spaces. Suppose that $\mathbf{F} : G \supset U \to H$ is Fréchet differentiable in $U$, that the derivative of $\mathbf{F}$ at $\Phi \in U$ is invertible, and $\mathbf{F}[\Phi] = 0$ uniquely. Then, there exists a neighbourhood $V \subset H$ of $0$ small enough such that*

$$\mathbf{F}[\mathcal{N}] \longrightarrow 0 \implies \mathcal{N} \longrightarrow \Phi.$$

Note that the existence of the Frechet derivative of $\mathbf{F}$ (denoted $\mathrm{D}\mathbf{F}$) is necessary if gradient descent or related methods are to work. As such, only the assumption that $\mathrm{D}\mathbf{F}^{-1}$ exists within some open neighbourhood of $\Phi$ is imposing additional structure. Theorem 1 is a powerful result backing the wisdom of using NN DE solvers. However, the same assumptions allow us to go much further:

**Theorem 2.** *Under the same assumptions as Theorem 1, we have*

$$\|\Phi_{\epsilon_1}\| = \mathcal{O}\left(\|\mathbf{F}[\mathcal{N}]\|\right).$$

A common observation made for NN DE solvers is that $\mathbb{E}\left[\|\mathbf{F}[\mathcal{N}]\|\right] \propto \mathbb{E}\left[\|\Phi_{\epsilon_1}\|\right]$. We codify that as a theorem over a relaxed class of assumptions on $\mathbf{F}$ as Theorem 2, validating the intuitive wisdom of using the $\mathcal{L}$ as a surrogate for error: especially when the associated constant of proportionality for the Lipschitz continuous $\mathbf{F}$ is of order unity. Some famous examples where Theorems 1 and 2 are valid are non-degenerate Hamiltonian systems, nonlinear Poisson-Boltzmann equation, Heat and Poisson's equation with homogeneous boundary conditions, etc. Our assumptions even guarantee an exponentially convergent optimization phase *somewhere* at *some rate* in $G$ (see Lemma 3 in Appendix B). However, we can go even further and explicitly estimate that rate with one additional assumption:

**Theorem 3.** *Let $G, H$ be Hilbert spaces, $U \subset G$ an open subset, and $\mathcal{N}(t)$ be the NN model after $t$ iterations. Assume $\mathbf{F} \in \mathscr{C}^2(U; H)$, and that $\mathrm{D}\mathbf{F}[\Phi]$ is invertible at $\Phi$. Then for every $\epsilon > 0$, there exists an $R > 0$, s.t. for all initial conditions $\mathcal{N}(0) \in B_R(\Phi) \subset U$, the gradient descent equation*

$$\dot{\mathcal{N}}(t) = -\nabla\mathcal{L}(\mathcal{N}(t))$$

*has a solution that satisfies*

$$\|\mathcal{N}(t) - \Phi\| \leq e^{-\frac{(1-\epsilon)\sigma_{\min}}{2}t}\|\mathcal{N}(0) - \Phi\|$$

*where*

$$\sigma_{\min} := \inf_{\mathcal{N} \in G \setminus \{0\}} \frac{\|\mathrm{D}\mathbf{F}[\Phi]\mathcal{N}\|^2}{\|\mathcal{N}\|^2} = \inf \mathrm{Spec}\left((\mathrm{D}\mathbf{F}[\Phi])^{\dagger}\mathrm{D}\mathbf{F}[\Phi]\right) > 0.$$

$\sigma_{\min} > 0$ implies exponential convergence at least at that rate is always possible. However, while we mitigate the concerns raised in the first two observations, Theorems 1 - 3 still only weakly/globally describe $\Phi_{\epsilon_1}$ over $\mathcal{D}$. We can obtain stronger estimates on $\|\Phi_{\epsilon_1}\|$ by assuming structural information on the $\Phi$ dependent terms $\mathbf{L}, \mathbf{N}$ and on the interactions they might have with each other.

In particular, assume that $\mathrm{D}\mathbf{N}$ is a positive (or negative) definite operator (such an assumption is often taken to avoid degenerate systems). Define $\mathcal{N}_s = \Phi - s\Phi_{\epsilon_1}$. Further, let:

$$H_{\min} := \inf_{s \in [0,1]} \inf \left\{|\lambda| \,\big|\, \lambda \in \mathrm{Spec}\left(\mathrm{D}\mathbf{N}[\mathcal{N}_s]\right)\right\}$$

Let $F_{\max}$ be the maximum of $\|\mathbf{F}[\mathcal{N}]\|$ over $\mathcal{D}$ for some model $\mathcal{N}$ that has finished optimizing. We then have the following inequality on $\|\Phi_{\epsilon_1}\|$ (see Appendix C for assumptions and proof):

$$\|\Phi_{\epsilon_1}(\mathbf{x})\| \leq \frac{F_{\max}}{H_{\min}} \qquad \mathbf{x} \in \mathcal{D} \tag{4}$$

A variation of Inequality 4 in Appendix C showcases how NN DE solver design can encode other knowledge/guesses on the system's behavior. Those can be used to fine-tune the assumptions that go into obtaining the presented inequality and lead us to modifications needed on the bound.

However, even with the stronger inequalities on $\|\Phi_{\epsilon_1}\|$ (which might still give severe overestimates, if $H_{\min}$ is not properly controlled for), we don't yet have information regarding pointwise behavior of $\Phi_{\epsilon_1}$ over $\mathcal{D} - \partial\mathcal{D}$. There is a scarcity of work on algorithms that can explicitly estimate $\Phi_{\epsilon_1}$ over $\mathcal{D} - \partial\mathcal{D}$, using only the information available to the model $\mathcal{N}$. More often than not, precise error estimations rely on actual knowledge of $\Phi$ over entirety of $\mathcal{D}$. In the next section, we shall provide an unsupervised algorithm to estimate $\Phi_{\epsilon_1}$ over $\mathcal{D} - \partial\mathcal{D}$, without knowledge of $\Phi$.

This endeavor, combined with the third observation will directly lead to another useful application. For any nontrivial DE, it is improbable that $\mathcal{N}$ over $\mathcal{D}$ will be the exact solution once a stable minima(s) of optimization is reached (stability is meant here in the sense of a converged model that is unlikely to find a significantly better minima in a *reasonable* amount of iterations). Indeed, it is demonstrably impossible if the DE does not admit closed form solutions, since $\mathcal{N}$ is always a closed form expression. Any further training would not provide any meaningful gains in accuracy.

However, in that regime, even a moderately good estimate $\mathcal{N}_{\epsilon_1}$ for $\Phi_{\epsilon_1}$ would be immediately useful: we could simply use $\mathcal{N} + \mathcal{N}_{\epsilon_1}$ as a better model (albeit, at additional optimization/model memory costs). Finally, just like Theorems 1 - 3 remove the ambiguity associated with standard NN DE solvers, we will find similar results reliably eliminate such ambiguities vis a vis the error model $\mathcal{N}_{\epsilon_1}$.

## 4 ERROR ESTIMATION AND CORRECTION

We begin by noticing the following:

$$\mathbf{F}[\mathcal{N}] = \mathbf{L}[\mathcal{N}] + \mathbf{N}[\mathcal{N}] + \mathbf{C} = -\mathbf{L}[\Phi_{\epsilon_1}] - \mathbf{N}'[\mathcal{N}, \Phi_{\epsilon_1}] \tag{5}$$

where $\mathbf{N}'[\mathcal{N}, \Phi_{\epsilon_1}] := \mathbf{N}[\mathcal{N} + \Phi_{\epsilon_1}] - \mathbf{N}[\mathcal{N}]$. Thus, we have a new equation and its operator $\mathbf{F}_1$

$$\mathbf{F}_1[\Phi_{\epsilon_1}] = \mathbf{F}[\mathcal{N}] + \mathbf{L}[\Phi_{\epsilon_1}] + \mathbf{N}'[\mathcal{N}, \Phi_{\epsilon_1}] = 0 \tag{6}$$

with $\Phi_{\epsilon_1}$ as the only unknown quantity. The uniqueness of $\Phi$ implies that if $\mathcal{N}$ is fixed, then $\Phi_{\epsilon_1}$ is a unique solution to the DE given by Eq. 6, and we have [1]

$$\mathbf{F}_1[\Phi_{\epsilon_1}] = \mathbf{F}[\mathcal{N} + \Phi_{\epsilon_1}] = 0$$

Note that $\mathbf{N}'$ is obtained such that terms explicitly dependent on $\Phi$ don't appear in Eq. 5: that is what allows tight bounds on $\Phi_{\epsilon_1}$ in (10), without depending upon explicit knowledge of $\Phi$ (Inequality 4 is a strong generalization of that result). As a motivating example, consider the cases when the nonlinearities in $\mathbf{F}$ are degree 2 polynomials: $\mathbf{N}[\;] = [\;] \cdot [\;]$. In that case, we have:

$$\mathbf{F}[\mathcal{N}] + \mathbf{L}[\Phi_{\epsilon_1}] + \mathbf{N}'[\mathcal{N}, \Phi_{\epsilon_1}] = 0 : \quad \mathbf{N}'[\mathcal{N}, \Phi_{\epsilon_1}] = (\Phi_{\epsilon_1} \cdot \Phi_{\epsilon_1} + \mathcal{N} \cdot \Phi_{\epsilon_1} + \Phi_{\epsilon_1} \cdot \mathcal{N})$$

The transformation technique itself (algebraic manipulations, Taylor expansions, etc) is not important: we only need it to be such that $\mathbf{F}[\mathcal{N}], \Phi_{\epsilon_1}$ appear in relation to each other over all of $\mathcal{D}$. Indeed, we are always able to directly leverage the unsimplified $\mathbf{N}[\mathcal{N}] - \mathbf{N}[\mathcal{N} + \Phi_{\epsilon_1}]$ form of $\mathbf{N}'$ (or even the $\mathbf{F}[\mathcal{N} + \Phi_{\epsilon_1}] = 0$ equation) [2] for optimization purposes in the proposed Algorithm 1.

---

[1] Note that $\mathbf{F}_1[\mathcal{N}_{\epsilon_1}] = \mathbf{F}[\mathcal{N} + \mathcal{N}_{\epsilon_1}]$ for **any** appropriate mapping $\mathcal{N}_{\epsilon_1}$, and not just $\Phi_{\epsilon_1}$

[2] However, it is sometimes profitable to obtain an explicit, separable expression for $\mathbf{N}'[\mathcal{N}, \Phi_{\epsilon_1}]$. For example, whenever Taylor expansions can be used,

$$\mathbf{N}'[\mathcal{N}, \Phi_{\epsilon_1}] \equiv \mathcal{T}_1(\mathcal{N})\Phi_{\epsilon_1} + \frac{\Phi_{\epsilon_1}^\dagger \mathcal{T}_2(\mathcal{N})\Phi_{\epsilon_1}}{2!} + \dots$$

where $\mathcal{T}_i(\mathcal{N})$ are the $i^{\text{th}}$ order Taylor terms. These can be useful for analysis, since $\mathcal{N}, \Phi_{\epsilon_1}$ appear in multiplicatively separable terms in the new expressions ((10) used these forms to bound $\Phi_{\epsilon_1}$). However, we obtained Inequality 4 in a way that renders such transformations superfluous: our bounds stand under weaker assumptions.

---

**Algorithm 1** Unsupervised Estimation and Correction of $\Phi_{\epsilon_1}$

---
1: **procedure** ERRORCORRECT
2:     **initialize** NN DE solver $\mathcal{N} : \mathcal{D} \to \mathbb{R}$.
3:     **while** $\mathcal{N}$ not converged **do**                     $\triangleright$ Initial model $\mathcal{N}$ over $\mathcal{D}$
4:         Optimize $\mathcal{N}$ for a step using $\mathcal{L}$ (Eq. 2)
5:     **end while**
6:     Freeze the parameters of $\mathcal{N}$                   $\triangleright$ Fix $\mathcal{N}$ to start error analysis
7:     **initialize** Error estimation model $\mathcal{N}_{\epsilon_1}$.
8:     **while** $\mathcal{N}_{\epsilon_1}$ not converged **do**              $\triangleright$ Error estimation over $\mathcal{D}$
9:         Optimize $\mathcal{N}_{\epsilon_1}$ for a step using $\mathcal{L}_\epsilon$ (Eq. 8)
10:     **end while**
11:     **return** $\mathcal{N} + \mathcal{N}_{\epsilon_1}$                      $\triangleright$ Error correction over $\mathcal{D}$
12: **end procedure**

---

Eq. 6 immediately hints at a method to estimate $\Phi_{\epsilon_1}$: we use Eq. 6 with another network model, $\mathcal{N}_{\epsilon_1}$, in the manner we used $\mathcal{N}$ with Eq. 2, to obtain an approximation for $\Phi_{\epsilon_1}$. More precisely, once we have a converged model $\mathcal{N}$ that has saturated its capacity to model $\Phi$, we use:

$$\mathbf{F}_1[\mathcal{N}_{\epsilon_1}] = \mathbf{F}[\mathcal{N}] + \mathbf{L}[\mathcal{N}_{\epsilon_1}] + \mathbf{N}'[\mathcal{N}, \mathcal{N}_{\epsilon_1}] \tag{7}$$

to define a new loss function for the error model $\mathcal{N}_{\epsilon_1}$:

$$\mathcal{L}_{\epsilon_1} = \mathbb{E}_{x_1 \in \mathcal{D}}\left[\|\mathbf{F}_1[\mathcal{N}_{\epsilon_1}(x_1)]\|_p\right] + \mathbb{E}_{x_2 \in \partial\mathcal{D}}\left[\|\Phi_{\epsilon_1}(x_2) - \mathcal{N}_{\epsilon_1}(x_2)\|_p\right] \tag{8}$$

Alternatively, since the parameters of $\mathcal{N}$ are kept unchanged, we can use the equivalent form:

$$\mathcal{L}_{\epsilon_1} = \mathbb{E}_{x_1 \in \mathcal{D}}\left[\|\mathbf{F}[\mathcal{N}(x_1) + \mathcal{N}_{\epsilon_1}(x_1)]\|_p\right] + \mathbb{E}_{x_2 \in \partial\mathcal{D}}\left[\|\Phi(x_2) - [\mathcal{N}(x_2) + \mathcal{N}_{\epsilon_1}(x_2)]\|_p\right] \tag{9}$$

In problems with homogeneous constraint conditions, $\Phi(\partial\mathcal{D}) = 0$, we may replace the second term in Eq. 8 by $\mathbb{E}\left[\|\mathcal{N}_{\epsilon_1}(x_2) + \mathcal{N}(x_2)\|^p\right]$, since $\Phi_{\epsilon_1}(\partial\mathcal{D}) = 0 - \mathcal{N}(\partial\mathcal{D}) = -\mathcal{N}(\partial\mathcal{D})$.

The same reasoning underlying $\mathcal{L} = 0 \implies \mathcal{N} = \Phi$, allows that $\mathcal{L}_{\epsilon_1} = 0 \implies \mathcal{N}_{\epsilon_1} = \Phi_{\epsilon_1}$. Indeed, analogs of Theorems 1 - 3 and Inequality 4 are obtained directly, as stated in a generalized form in the next subsection. In particular, our knowledge of $\mathbf{F}[\mathcal{N}]$ allows us substantial control over how to estimate $\Phi_{\epsilon_1}$, whereas the knowledge of $\mathbf{F}_1[\mathcal{N}_{\epsilon_1}]$ (or alternatively $\mathbf{F}[\mathcal{N} + \mathcal{N}_{\epsilon_1}]$) allows us bounds on $\|\Phi_{\epsilon_1} - \mathcal{N}_{\epsilon_1}\|$. The ability to *internally* **and** *reliably* estimate $\Phi_{\epsilon_1}$ associated with $\mathcal{N}$ is not only useful for error analysis purposes, but also for refining $\mathcal{N}$. Algorithm 1 describes the procedure.

Note that the difficulty of estimating $\Phi$ and $\Phi_{\epsilon_1}$ should be roughly equivalent since $\Phi_{\epsilon_1}$ has at least the same regularity as $\Phi$. As such, the optimization costs of $\mathcal{N}_{\epsilon_1}$ per iteration will not be substantially larger than those of $\mathcal{N}$ (see discussion/experiments in the next sections).

Finally, note that Alg. 1 provides relative corrections to some model $\mathcal{N}$. We do not claim that only error correction can provide improvements in performance to the extent shown in Fig. 2. We only claim that error analysis and correction are useful and available tools for *any* model $\mathcal{N}$ that might have been obtained via some technique based on NN DE solvers. It should **always** be possible to error correct $\mathcal{N}$ using the strategy we are describing, as long as the associated assumptions are satisfied.

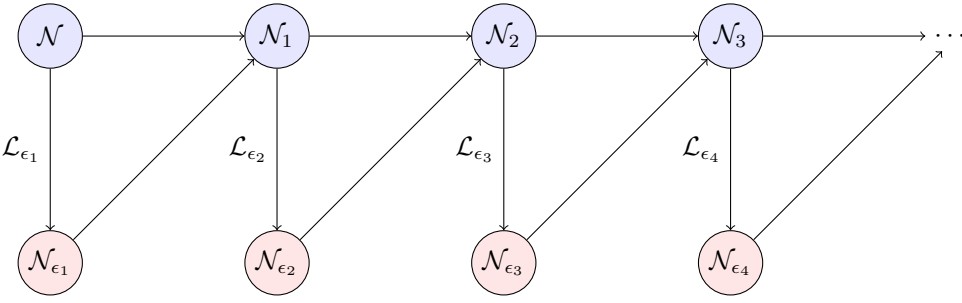

Figure 1: Graphical Model of the $i^{\text{th}}$ order error correction scheme. Each vertical pair of blue and red nodes denotes a specific *order* in the estimation and correction process.

### 4.1 HIGHER ORDER ERROR ESTIMATIONS AND CORRECTIONS

Consider some sequence of networks $\mathcal{N}_{\epsilon_1}, \mathcal{N}_{\epsilon_2}, ...., \mathcal{N}_{\epsilon_{i-1}}$, each modeling the errors $\Phi_{\epsilon_1}, \Phi_{\epsilon_2}, ...., \Phi_{\epsilon_i}$, associated with error corrected solution models $\mathcal{N}, \mathcal{N}_1, ..., \mathcal{N}_{i-1}$. Here, $\mathcal{N}_j = \mathcal{N} + \mathcal{N}_{\epsilon_1} + ... + \mathcal{N}_{\epsilon_j}$ is the $j^{\text{th}}$ order corrected model. Assume the $i-1$ order error correction model $\mathcal{N}_{\epsilon_{i-1}}$ has saturated its capacity to model $\Phi_{\epsilon_{i-1}}$. We can obtain a higher order corrected model $\mathcal{N}_i$ by setting up a new network $\mathcal{N}_{\epsilon_i}$ to estimate $\Phi_{\epsilon_i}$. The new network $\mathcal{N}_{\epsilon_i}$ is optimized using:

$$\mathbf{F}_i[\mathcal{N}_{\epsilon_i}] := \mathbf{F}[\mathcal{N}_{\epsilon_i} + \mathcal{N}_{i-1}] = \mathbf{F}_{i-1}[\mathcal{N}_{i-1}] + \mathbf{L}[\mathcal{N}_{\epsilon_i}] + \mathbf{N}[\mathcal{N}_{i-1} + \mathcal{N}_{\epsilon_i}] - \mathbf{N}[\mathcal{N}_{i-1}]$$

$$\mathcal{L}_{\epsilon_i} = \mathbb{E}_{x_1 \in \mathcal{D}}\Big[\|\mathbf{F}_i[\mathcal{N}_{\epsilon_i}(x_1)]\|_p\Big] + \mathbb{E}_{x_2 \in \partial\mathcal{D}}\Big[\|\Phi_{\epsilon_i}(x_2) - \mathcal{N}_{\epsilon_i}(x_2)\|_p\Big] \tag{10}$$

Figure 1 provides a visual representation of the recursive process involved in building $i^{\text{th}}$ order corrected models. Appendix A demonstrates how to arrive at these results.

We have direct analogs of Theorems 1, 2, and 3 for the $i^{\text{th}}$ order errors and error models as:

**Theorem 4.** *Under the assumptions above on* $\mathbf{F}$, *the analogous result holds for* $\mathbf{F}_i$ *for all* $i \in \mathbb{N}$, *i.e. we can choose a neighbourhood* $U_i \subset G$ *of 0 small enough such that if* $\mathcal{N}_{\epsilon_i} \in U_i$ *then*

$$\mathbf{F}_i[\mathcal{N}_{\epsilon_i}] \longrightarrow 0 \implies \mathcal{N}_{\epsilon_i} \longrightarrow \Phi_{\epsilon_i}.$$

*Furthermore,*

$$\|\Phi_{\epsilon_{i+1}}\| = \mathcal{O}\left(\|\mathbf{F}_i[\mathcal{N}_{\epsilon_i}]\|\right).$$

*Lastly, under gradient descent, initial condition* $\mathcal{N}_{\epsilon_i}(0)$ *has a solution that satisfies*

$$\|\mathcal{N}_{\epsilon_i}(t) - \Phi_{\epsilon_i}\| \leq e^{-\frac{(1-\epsilon)\sigma_{i_{\min}}}{2}t}\|\mathcal{N}_{\epsilon_i}(0) - \Phi_{\epsilon_i}\|$$

Inequality 4 becomes (under the same structural assumptions as Inequality 4):

$$\|\Phi_{\epsilon_i}\| \leq \frac{F_{i-1_{\max}}}{H_{\min}} \tag{11}$$

Algorithm 2 in the appendix provides the pseudo-code to setup higher order corrections.

## 5 WHY ERROR CORRECTION WORKS AND HOW WE CAN MAKE IT BETTER

NN DE solvers transform the problem of estimating $\Phi$ into one of finding an appropriate $\mathcal{N}$ in some suitably chosen functional space of NN models. The use of gradient descent type methods turns this into a stochastic dynamical process in that space.

The probability that a real network $\mathcal{N}$ will end up exactly modeling $\Phi$ within a finite number of iterations is 0 (assuming $\mathbf{F}$ is even moderately non-trivial). Note this is true without even considering that $\mathcal{N}$ is constrained to be a closed form model, whereas most non-trivial DEs don't have such solutions. As such, a real network $\mathcal{N}$ is doomed to languish in some inadequate minima(s) of optimization, no matter how much we optimize it. Its capacity to approximate $\Phi$ is necessarily limited and once that capacity is saturated, further training of $\mathcal{N}$ is a waste of computational resources.

Error correction expands that capacity for approximating $\Phi$, by introducing new trainable parameters (as components of $\mathcal{N}_{\epsilon_1}$) which exist exclusively for modeling the error $\Phi_{\epsilon_1} = \Phi - \mathcal{N}$. Since the parameters of the base network $\mathcal{N}$ are frozen, we have locked in the amount of performance already achieved on the base model. Further, since we assume error correction is being turned on when the base model's approximation capacity is effectively saturated (which is certain to happen), the gains we make with $\mathcal{N}_{\epsilon_1}$ are truly unachievable without it.

We have put forth a claim that error correction is not significantly more complex or resource hungry than the standard NN DE solver method: the reason being that $\Phi$ and $\Phi - \mathcal{N}$ are in the same functional space, have the same regularity, etc. Essentially, we meant that $\Phi_{\epsilon_1}$ is no more *complicated* than $\Phi$. However, it bears noting that modeling $\Phi_{\epsilon_1}$ using $\mathcal{N}_{\epsilon_1}$ can be a very different kind of problem than modeling $\Phi$ using $\mathcal{N}$. For example, $\Phi_{\epsilon_1}$ presumably lies in a different scale of magnitude than $\Phi$ and might present itself as a different kind of structure over $\mathcal{D}$ (for example, in one sense, it might be significantly more or less "oscillatory" than $\Phi$). Its optimization might also require its own set of hyper-parameters, algorithm choices, activations, etc.

This is where our control over the sampling of $\mathbf{F}$ (and $\mathbf{F}_1$) comes into play. Theorems 1 - 3 and Inequality 4 allow us to estimate both the scale and complexity of $\Phi_{\epsilon_1}$, relative to $\mathcal{N}$. The correlation expected between $\Phi_{\epsilon_1}$ and $\mathbf{F}[\mathcal{N}]$ means even crude analysis of $\mathbf{F}[\mathcal{N}]$ is enough to point us in the correct direction, vis a vis the scale and complexity we expect $\mathcal{N}_{\epsilon_1}$ to model.

For example, we can bound the expected scale of $\Phi_{\epsilon_1}$ by simply analysing the max of $\|\mathbf{F}[\mathcal{N}]\|$ over $\mathcal{D}$: the linear correlation result already gives us one way to weakly bound $\Phi_{\epsilon_1}$. The strong inequality 4 may allow for more precise control, when its assumptions are satisfied.

Similarly, over a domain $\mathcal{D}$, we could perform a FFT analysis of $\mathbf{F}[\mathcal{N}]$ and gauge how oscillatory $\mathcal{N}_{\epsilon_1}$ should be relative to $\mathcal{N}$ (since we have control over both $\mathcal{N}$ and $\mathbf{F}[\mathcal{N}]$ and we expect $\Phi_{\epsilon_1}$ to be linearly correlated to $\mathbf{F}[\mathcal{N}]$). Alternatively, we could also analyse the distribution of $\|\mathbf{F}[\mathcal{N}]\|$ over $\mathcal{D}$ to understand where the equation is being badly solved, and switch from uniform to targeted sampling over those subdomains. Other methods of estimating relative complexity of features should also be possible, since we have complete access to the behavior of $\mathbf{F}[\mathcal{N}]$ and $\mathbf{F}_1[\mathcal{N}_{\epsilon_1}]$.

The code we provide allows the user to supply their own estimates for relative scale and complexity of $\mathcal{N}_{\epsilon_1}$ wrt $\mathcal{N}$, while keeping our preferred defaults. We note again, that in no way is prior knowledge of $\Phi$ assumed: only the information already known before or computed using the NN DE solver is being used. Domain knowledge of an expert building an NN DE solver for a system of their interest is thus rewarded during error analysis and correction.

Finally, note that error correction inherits one major issue prevalent within machine learning: the ambiguity in determining if a real model $\mathcal{N}$ has reached (or nearly reached) its capacity for effectively modelling $\Phi$. The issue is caused because of lack of work on *a priori* or *in situ* convergence guarantees. It is extremely tough to gauge whether a loss trajectory has flattened due to saturation in modeling capacity or it is simply in a non-trivial region of dynamics from which it could escape at any time. If error correction is attempted while the base model itself is fully capable of providing significant increases in accuracy, the additional resources will simply be wasted (and may even prove to be counter-productive). Indeed, in our numerical experiments, we found that convergence of the base model $\mathcal{N}$ was a practical pre-requisite for significant gains via error correction.

However, in our proposed approach, the issue is still somewhat mitigated. Error correction can not cost significantly more than the standard methods (it can at worst be 2X expensive). Since it can lead to very significant improvements in accuracy when used at the appropriate time, it can always be used as an unsupervised, post-processing/refinement routine without significant risks.

## 6 Numerical Experiments

Let us exemplify the above discussion with an example DE that contains terms with linear, non-linear, and no dependence on the solution. In particular, we investigate a variant of the nonlinear Poisson-Boltzmann equation (nPBE), which serves as an important tool for the study of electrostatic interactions with widespread applications in biology and medicine (14). We will choose $\mathbf{L}[\,] = -\Delta[\,], \mathbf{N}[\,] \equiv \sinh[\,], \mathbf{C} \equiv g$, where $g$ is some independent function/term. Thus, we define:

$$\mathbf{F}[\,] \equiv -\Delta[\,] + \sinh[\,] + g \tag{12}$$

For some solution attempt $\mathcal{N} : \mathbb{R}^d \to \mathbb{R}$ and associated error $\Phi_{\epsilon_1} = \Phi - \mathcal{N}$, we get

$$\mathbf{F}[\mathcal{N}] = \Delta[\Phi_{\epsilon_1}] + \sinh[\mathcal{N}] - \sinh[\mathcal{N} + \Phi_{\epsilon_1}] \tag{13}$$

Thus, replacing $\Phi_{\epsilon_1}$ with an error model $\mathcal{N}_{\epsilon_1}$, we get the residual equation for the new loss as:

$$\mathbf{F}_1[\mathcal{N}_{\epsilon_1}] := \mathbf{F}[\mathcal{N} + \mathcal{N}_{\epsilon_1}] = \mathbf{F}[\mathcal{N}] - \Delta[\mathcal{N}_{\epsilon_1}] + \sinh[\mathcal{N}](\cosh[\mathcal{N}_{\epsilon_1}] - 1) + \cosh[\mathcal{N}]\sinh[\mathcal{N}_{\epsilon_1}] \tag{14}$$

Now, to setup $\mathcal{N}_{\epsilon_1}$ as the error NN DE solver (after halting the optimization of $\mathcal{N}$), we can use Eq. 14 to obtain an approximation of $\Phi_{\epsilon_1}$. We not only get an estimate for $\Phi_{\epsilon_1}$, we also get to use $\mathcal{N}_{\epsilon_1}$ as a correction term, while retaining the advantages that make $\mathcal{N}$ a lucrative option.

### 6.1 Error correction results

We choose $\mathcal{D} = [-\pi, \pi]^d \subset \mathbb{R}^d$, and randomly sample points from this domain as training data. The exact form of our chosen DE, considered over $\mathcal{D}$ with solution $\Phi : \mathcal{D} \to \mathbb{R}$ and homogeneous

boundary conditions $\Phi(\partial \mathcal{D}) = 0$, is:

$$\mathbf{F}[\Phi] = -\Delta[\Phi] + \sinh[\Phi] + g = 0 \tag{15}$$

$$g(\mathbf{x}) = -\omega^2 d \left[\sin(\omega x_1) ... \sin(\omega x_d)\right] - \sinh(\sin(\omega x_1) ... \sin(\omega x_d)) \qquad \mathbf{x} \equiv \{x_1, ..., x_d\} \in \mathcal{D}$$

The solution to this DE, which showcases significant scale and feature variation over $\mathcal{D}$ (Fig. 3), is:

$$\Phi(\mathbf{x}) = \sin(\omega x_1) ... \sin(\omega x_d) \qquad \mathbf{x} \in \mathcal{D} \tag{16}$$

Note that the solvers never see this solution, and only have access to their own parameters and the operator $\mathbf{F}$ during optimization. We only use the exact solution to verify that our claims hold true.

Given a network $\mathcal{N}$ to model the true solution $\Phi$, we optimize $\mathcal{N}$ using the mean squared loss $\mathcal{L}$:

$$\mathcal{L}(\mathcal{N}) = \mathbb{E}_{x_1 \sim \mathcal{D}} \left[\| -\Delta(\mathcal{N}(x_1)) + \sinh[\mathcal{N}(x_1)] + g(x_1)\|^2\right] + \mathbb{E}_{x_2 \sim \partial \mathcal{D}} \left[\|\mathcal{N}(x_2)\|^2\right] \tag{17}$$

The error correction model $\mathcal{N}_{\epsilon_1} : \mathcal{D} \to \mathbb{R}$ is effectively optimized using the following loss:

$$\mathcal{L}_{\epsilon_1} = \mathbb{E}_{x_1 \sim \mathcal{D}} \left[\|\mathbf{F}_1[\mathcal{N}_{\epsilon_1}(x_1)]\|^2\right] + \mathbb{E}_{x_2 \sim \partial \mathcal{D}} \left[\|\mathcal{N}(x_2) + \mathcal{N}_{\epsilon_1}(x_2)\|^2\right] \tag{18}$$

As discussed before, $\mathcal{L} \to 0 \implies \mathbf{F}[\mathcal{N}(\mathbf{x})] \to 0$, over $\mathbf{x} \in \mathcal{D}$, which implies $\mathcal{N} \to \Phi$. Similarly, $\mathcal{L}_{\epsilon_1} \to 0 \implies \mathbf{F}_1[\mathcal{N}_{\epsilon_1}(\mathbf{x})] \to 0$, which implies $\mathcal{N}_{\epsilon_1} \to \Phi_{\epsilon_1}$. We use Eq. 16 to benchmark efficacy of our proposed algorithm for $(\omega, d)$ choices of $(5, 2)$ and $(1, 4)$. The goal of the experiments was to figure out what impact error correction would have, if used in lieu of the standard approach, while running for an equivalent number of training iterations. As such, choices on width, depth, architecture, etc were kept the same between $\mathcal{N}$ and $\mathcal{N}_{\epsilon_1}$. The experiments also demonstrate the effects of invoking error correction at different stages of training (Fig. 2).

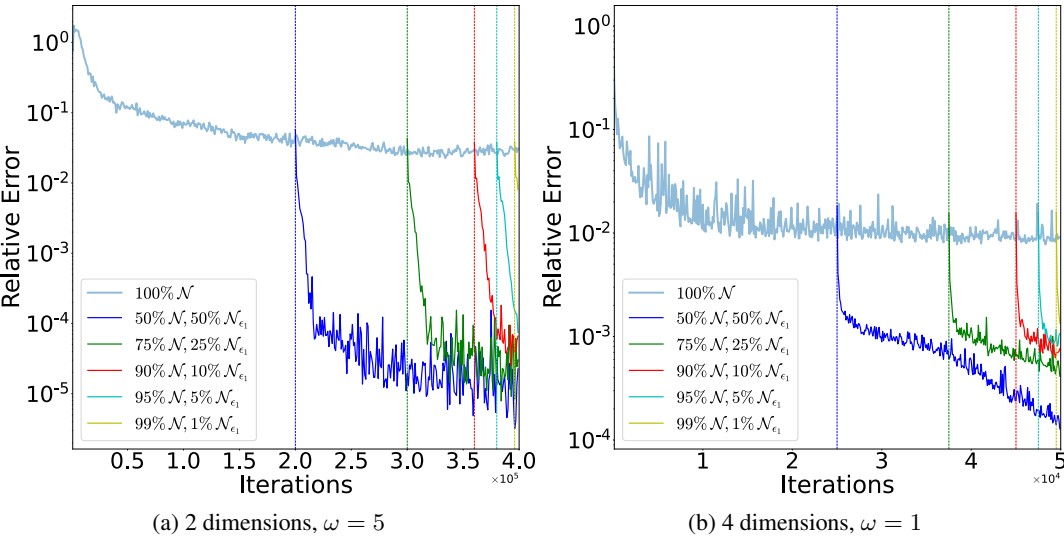

(a) 2 dimensions, $\omega = 5$         (b) 4 dimensions, $\omega = 1$

Figure 2: Relative errors for a single order correction on nPBE. Legend labels indicate duration of training $\mathcal{N}$, and duration of training $\mathcal{N}_{\epsilon_1}$ for correction respectively.

To present and analyse our results, we define a computational cost metric, $\tau$: (time per iteration), and a computational performance metric, Relative Error (RE):

$$\text{Relative Error} := \left( \sum_{k=1}^{K} [\Phi(\mathbf{x}_k) - \mathcal{N}_i(\mathbf{x}_k)]^2 \Big/ \sum_{k=1}^{K} [\Phi(\mathbf{x}_k)]^2 \right)^{\frac{1}{2}},$$

where $\mathcal{N}_i : \mathcal{D} \to \mathbb{R}$ is the $i^{\text{th}}$ order corrected, NN DE based approximation for $\Phi$ over $\mathcal{D}$. $K$ represents the number of sampled points in $\mathcal{D}$ for the calculation. In Fig. 2, we plot the relative error dynamics of the various choices made. Notice the very immediate gains made by error correction.

Table 1: **Performance comparison across different systems and optimization strategies**

| System (Adapted Codebase) | $\tau$ ($10^{-2}$ s) (Std, EC) | Rel. Error ($10^{-4}$) (Standard, EC) | ROI (EC) | ROI (Size Ablation) | ROI (Time Ablation) |
|---|---|---|---|---|---|
| HH (exact IC HNN (10)) | (1.37, 1.63) | (3.08, 0.18) | **14.4** | 1.38 | 0.81 |
| nOsc (exact IC HNN (10)) | (1.39, 1.68) | (1.73, 0.02) | **71.6** | 1.12 | 0.83 |
| nPBE (5, 2) (PINN (2)) | (8.6, 11.8) | (436, 0.37) | **861** | 1.65 | 0.77 |
| nPBE (1, 4) (PINN (2)) | (72.1, 89.8) | (87.3, 1.87) | **37.5** | 1.23 | 0.79 |

We also investigate alternative scenarios, where the slightly higher consumption of resources is used in other ways, to quantify ablation possibilities. We conduct two sets of examples. For size ablation experiments, we allow $\mathcal{N}$ to be proportionally larger, while training it for the same number of training iterations (say $\mathcal{T}$). For time ablation experiments, we allow the standard algorithms a proportionally higher number of iterations. We define the **Return on Investment** or **ROI** of each choice $c =$ (error correction, size, or time ablation) as:

$$\text{ROI}(c) = \frac{\tau_{\text{Std}}}{\tau(\text{c})} \times \frac{\text{RE}(\text{c})}{\text{RE}_{\text{Std}}} \times \frac{\mathcal{T}_{\text{Std}}}{\mathcal{T}(c)}$$

As Table 1 shows, error correction consistently provides the best ROI.

The associated codebase with this work provides numerical experiments on some other nonlinear/chaotic DEs (it is designed to be compatible with any DE operator with permissible Frechet derivatives and allow any finite order of correction). The results on those systems are also tabulated in Table 1 (first order error correction was injected at 50% of the run-time, for all other examples). All results are in line with our expectations. The additional systems are described in appendix D.

## 7 CONCLUSIONS

The surging popularity of NN DE solvers presents exciting possibilities in many scientific fields: their capacity to sidestep the curse of dimensionality (15), general advances in computing/GPU power, and their easy interpret-ability make for a powerful combo. As such, it is important that these solution models be capable of validation over domains where true solutions are not available. Our work proposes theorems/methods that fix this deficiency for many NN DE solvers. Summarily, NN model errors are unambiguously estimable, and profitably so, if the assumptions of Theorem 1 - 3 hold.

For systems where the assumptions of Theorems 1 - 3 do not hold, existing work still leads us to expect that NN DE solvers (**and** thus, error correction) will work. However, for those systems, error correction converts the ambiguity associated with the models $\mathcal{N}$ into the ambiguity associated with the estimate $\mathcal{N}_{\epsilon_1}$. These unreliable estimates of $\Phi_{\epsilon_1}$ can lead to more accurate models, but our work does not rigorously predict when or how this happens. However, since this ambiguity is present **for all** NN DE solvers prior to our work, unsupervised error analysis and correction can only be an improvement upon the existing situation (even when it is unreliable). Future work will focus on extending our results rigorously to more systems, so that even larger classes of DEs may come within the remit of **reliable** error estimation and correction.

Lastly, note that the suggested ideas are not radically different than those that already exist for classical numerical methods: higher order corrections are about as old as the field itself. By pairing the many significant advantages of modern NNs with those old ideas, we simply hope to have presented a blueprint that will be useful for a wide class of scientific problems that are being tackled using NN DE solvers.

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

## A  SETTING UP HIGHER ORDER CORRECTIONS

Using Eq. equation 7, we are able to define new loss $\mathcal{L}_{\epsilon_1}$ using $\mathbf{F}_1[\mathcal{N}_{\epsilon_1}]$ (or equivalently, $\mathbf{F}[\mathcal{N} + \mathcal{N}_{\epsilon_1}]$). Define $\Phi_{\epsilon_2} := \Phi_{\epsilon_1} - \mathcal{N}_{\epsilon_1} = \Phi - (\mathcal{N} + \mathcal{N}_{\epsilon_1})$, and use the same trick used to derive Eq. 5 to realize that,

$$\begin{aligned}
\mathbf{F}_1[\mathcal{N} + \mathcal{N}_{\epsilon_1}] &= \mathbf{F}[\mathcal{N}] + \mathbf{L}[\Phi_{\epsilon_1} - \Phi_{\epsilon_2}] + \mathbf{N}[\mathcal{N} + \mathcal{N}_{\epsilon_1}] - \mathbf{N}[\mathcal{N}] + (\mathbf{N}[\mathcal{N} + \Phi_{\epsilon_1}] - \mathbf{N}[\mathcal{N} + \Phi_{\epsilon_1}]) \\
&= -\mathbf{L}[\Phi_{\epsilon_2}] + \mathbf{N}[\mathcal{N} + \mathcal{N}_{\epsilon_1}] - \mathbf{N}[\mathcal{N} + \Phi_{\epsilon_1}] \\
&= -\mathbf{L}[\Phi_{\epsilon_2}] + \mathbf{N}[\mathcal{N} + \mathcal{N}_{\epsilon_1}] - \mathbf{N}[\mathcal{N} + \mathcal{N}_{\epsilon_1} + \Phi_{\epsilon_2}]
\end{aligned} \tag{19}$$

Let us define the $1^{st}$ order corrected model as $\mathcal{N}_1 := \mathcal{N} + \mathcal{N}_{\epsilon_1}$, and with a slight abuse of notation, $\mathbf{F}_1[\mathcal{N}_1] := \mathbf{F}[\mathcal{N} + \mathcal{N}_{\epsilon_1}]$. Thus, we may rewrite Eq. 19 as:

$$\mathbf{F}_1[\mathcal{N}_1] + \mathbf{L}[\Phi_{\epsilon_2}] + \mathbf{N}[\mathcal{N}_1 + \Phi_{\epsilon_2}] - \mathbf{N}[\mathcal{N}_1] = 0$$

Note the functional similarity to Eq. 6. Further, to setup a $2^{nd}$ order correction estimate $\mathcal{N}_{\epsilon_2}$ for $\Phi_{\epsilon_2}$, we need to minimize over the following residual, similar to Eq. 7:

$$\mathbf{F}_2[\mathcal{N}_{\epsilon_2}] = \mathbf{F}_1[\mathcal{N}_1] + \mathbf{L}[\mathcal{N}_{\epsilon_2}] + \mathbf{N}[\mathcal{N}_1 + \mathcal{N}_{\epsilon_2}] - \mathbf{N}[\mathcal{N}_1]$$

In general, given an $(i-1)^{th}$ order corrected model $\mathcal{N}_{i-1}$ using $\mathcal{N}_{i-1} = \mathcal{N} + \mathcal{N}_{\epsilon_1} + \mathcal{N}_{\epsilon_2} + ..... + \mathcal{N}_{\epsilon_{i-1}}$, we can define $\Phi_{\epsilon_i} := \Phi - \mathcal{N}_{i-1}$ and $\mathbf{F}_{i-1}[\mathcal{N}_{i-1}]$ as the corresponding residual for the preceding equations. Then, $\Phi_{\epsilon_i}$ is the unique solution to the following equation

$$\mathbf{F}_{i-1}[\mathcal{N}_{i-1}] + \mathbf{L}[\Phi_{\epsilon_i}] + \mathbf{N}[\mathcal{N}_{i-1} + \Phi_{\epsilon_i}] - \mathbf{N}[\mathcal{N}_{i-1}] = 0 \qquad (20)$$

The error can be estimated using a differential equation solver (e.g. a NN) with residual:

$$\mathbf{F}_i[\mathcal{N}_{\epsilon_i}] := \mathbf{F}[\mathcal{N}_{\epsilon_i} + \mathcal{N}_{i-1}] = \mathbf{F}_{i-1}[\mathcal{N}_{i-1}] + \mathbf{L}[\mathcal{N}_{\epsilon_i}] + \mathbf{N}[\mathcal{N}_{i-1} + \mathcal{N}_{\epsilon_i}] - \mathbf{N}[\mathcal{N}_{i-1}] \qquad (21)$$

Figure 1 depicts the extension of the error estimation and correction framework as a graphical model. Notice that the initial solver estimator and the error estimators may be pre-fetched before carrying out the optimization steps. Since $\Phi_{\epsilon_i} = \Phi - \mathcal{N}_{i-1}$ is always known exactly over $\partial\mathcal{D}$ for all $i$, we can optimize the $i^{th}$ order error model $\mathcal{N}_{\epsilon_i}$ using $\mathcal{L}_{\epsilon_i}$:

$$\mathcal{L}_{\epsilon_i} = \mathbb{E}_{x_1 \sim \mathcal{D}}\Big[\|\mathbf{F}_i[\mathcal{N}_{\epsilon_i}(x_1)]\|^p\Big] + \mathbb{E}_{x_2 \sim \partial\mathcal{D}}\Big[\|\Phi_{\epsilon_i}(x_2) - \mathcal{N}_{\epsilon_i}(x_2)\|^p\Big] \qquad (22)$$

Algorithm 2 demonstrates the procedure for higher order estimation and correction. The uniqueness of $\Phi$ as a solution to $\mathbf{F}[\ ]$ also implies the uniqueness of $\Phi_{\epsilon_i}$ as a solution to $\mathbf{F}_i[\ ]$. Unpacking the definition of $\mathbf{F}_i$ and $\mathcal{N}_{i-1}$ we find that

$$\mathbf{F}_i[\Phi_{\epsilon_i} + \mathcal{N}_{i-1}] = \mathbf{F}[\Phi - \mathcal{N}_{i-1} + \mathcal{N}_{i-1}] = \mathbf{F}[\Phi] = 0$$

and therefore the solution to this equation is still unique and convergence of $\mathcal{L}_{\epsilon_i} \to 0$ implies $\mathbf{F}_i \to 0$, and thus also $\mathcal{N}_{\epsilon_i} \to \Phi_{\epsilon_i}$. Theorem 3 follows naturally from Theorems 1 and 2. Inequality 11 follows naturally from Inequality 4. Algorithm 2 describes the proposed implementation.

---

**Algorithm 2** Internal Error Estimation and Correction to Order $m$

---

1: **procedure** ERRORCORRECT($m$)
2:     **initialize** NN DE solver $\mathcal{N} : \mathcal{D} \to \mathbb{R}$
3:     **initialize** Error estimators $\{\mathcal{N}_{\epsilon_i}\}_{i=1}^m$
4:     Train $\mathcal{N}$, and define $\hat{\mathcal{N}} := \mathcal{N}$                           $\triangleright \hat{\mathcal{N}}$ serves as a dummy variable
5:     **for** $i = 1, \ldots, m$ **do**
6:         Train $\mathcal{N}_{\epsilon_i}$ until loss converges
7:         Save parameter states of $\mathcal{N}_{\epsilon_i}$
8:         $\hat{\mathcal{N}} := \hat{\mathcal{N}} + \mathcal{N}_{\epsilon_i}$
9:         Freeze parameters of $\hat{\mathcal{N}}$
10:     **end for**
11:     **return** $\mathcal{N} + \sum_{i=1}^m \mathcal{N}_{\epsilon_i}$                      $\triangleright$ Equivalently, return $\hat{\mathcal{N}}$
12: **end procedure**

---

# B   THEOREMS ON THE RELATIONSHIPS BETWEEN $\Phi_{\epsilon_1}, \mathbf{F}[\mathcal{N}]$, AND $\mathcal{L}$

This work is centered around three theorems. We begin this section by summarizing and contextualizing those results.

Theorem 1 verifies the intuitive expectation that $\mathbf{F}[\mathcal{N}] \to 0$ implies the fitness of $\mathcal{N}$ as an approximation of $\Phi$. We do this by taking advantage of the uniqueness of $\Phi$ as a solution, alongside the fact that any gradient descent method can only be successful, iff the gradient itself exists in a well defined sense.

Theorem 2 naturally follows as a consequence of Theorem 1. It adds a strong *a priori* expectation we should have from NN DE solvers that satisfy the assumptions of Theorem 1 and tells us that their errors should be quantifiable in some sense, if information about $\mathbf{F}[\mathcal{N}]$ is sampled (which is always possible up to the desired resolution). Additional structure leads us to stronger quantification capabilities (surmised in Inequalities 4 and 11 and proved in appendix C).

Finally, assume a mapping $\mathbf{F} : G \supset U \to H$ between two Hilbert spaces, e.g. representing a non-linear PDE mapping between function spaces. Further, assume it is twice differentiable and at the solution $\Phi$, satisfying $\mathbf{F}(\Phi) = 0$, $\mathrm{D}\mathbf{F}[\Phi]$, is an invertible linear map. Then, the gradient descent procedure guarantees convergence at a rate

$$\|\mathcal{N}(t) - \Phi\| \leq e^{-\frac{(1-\epsilon)\sigma_{\min}}{2}t}\|\mathcal{N}(0) - \Phi\|$$

where $\epsilon > 0$ can be chosen sufficiently small if $\|\mathcal{N}(0) - \Phi\| \leq R$, where $R$ depends on $\epsilon$. $\sigma_{\min}$ here is given by the minimum of the spectrum of $(\mathrm{D}\mathbf{F}[\Phi])^{\dagger}\mathrm{D}\mathbf{F}[\Phi]$ which is strictly positive as $\mathrm{D}\mathbf{F}[\Phi]$ is invertible. Theorem 3 guarantees that exponentially converging NN models for $\Phi$ exist somewhere in $G$, and that gradient descent will allow us that level of performance. We need only find some initial model from that region.

Together, the three theorems give us an idealized framework for describing NN DE solvers under iterative optimization. Real NN DE solvers differ from this idealized framework in two ways: their optimization is a discrete process and the subspace within which real NN models lie (say $G_M$, where $M$ is number of parameters in our model), while being capable of coming arbitrarily close to $\Phi$, seldom contains it. The real optimization is a discrete approximation of the ideal trajectory: further, the empirical trajectory is actually a projection of that discrete trajectory onto $G_M$.

Note that for finite dimensional approximation of the problem on a subspace $G_M \subset G$ of dimension $M$, the corresponding constant $\sigma_{\min}^{G_M}$ is greater or equal to $\sigma_{\min}$. Furthermore, if $G_M \subset G_m$ then $\sigma_{\min}^{G_M} \geq \sigma_{\min}^{G_m}$. Thus, one can expect the error correction procedure to allow for an exponential improvement at each step, iff $\sigma_{\min} > 0$, i.e. exactly in the cases when $\mathrm{D}\mathbf{F}[\Phi]$ is invertible.

Finally, Theorem 4 follows trivially from the definitions of $\mathbf{F}_i, \mathcal{N}_{i-1}, \Phi_{\epsilon_i}$, and Theorems 1 - 3. We now move to rigorously prove Theorems 1 - 3 below.

## B.1 INVERSE FUNCTION THEOREM AND THEOREMS 1 AND 2

We quickly summarize the Banach space version of the inverse function theorem that allows us to establish that $\mathbf{F}[\mathcal{N}] \to 0$ implies $\mathcal{N} \to \Phi$ (for further details see for example (16)). We also establish another lemma we will use to estimate the convergence rate of NN DE solvers.

Let $G$ and $H$ be two Banach spaces, and $U \subset G$ an open subset. A continuous map between the two Banach spaces $\mathbf{F} : G \supset U \to H$ is said to be Fréchet differentiable at a point $x \in U$ iff there exists a linear bounded operator $L_x : G \to H$ such that

$$\|\mathbf{F}(y) - \mathbf{F}(x) - L_x(y - x)\|_H = o(\|y - x\|_G).$$

If the map $x \mapsto \mathrm{D}\mathbf{F}[x] = L_x$ is continuous, then one says that $\mathbf{F} \in \mathscr{C}^1(U; H)$. Analogously $\mathscr{C}^2(U; H)$ denote the functions which are twice differentiable, and if $H = \mathbb{R}$ we drop $H$ from the notation.

**Lemma 1.** *Let $\mathbf{F} : U \to H$ be a $\mathscr{C}^1$-map. Suppose that there exists a point $x_0 \in U$ such that $\mathrm{D}\mathbf{F}(x_0)$ is an isomorphism (i.e. it has a continuous inverse). Then there exists a neighbourhood $V \subset H$ of $\mathbf{F}(x_0)$ and a $\mathscr{C}^1$ function $\mathbf{F}^{-1} : V \to G$ that is a local inverse of $\mathbf{F}$.*

*Proof.* See Theorem 5.2 of (16). □

**Lemma 2.** *Assume $\mathcal{L} \in \mathscr{C}^2(U)$. $\mathcal{L}$ is strongly convex at $\mathcal{N} \in U$, if for all $\mathcal{N}' \in G$*

$$\mathrm{D}^2\mathcal{L}[\mathcal{N}](\mathcal{N}', \mathcal{N}') \geq \mu\|\mathcal{N}'\|^2 .$$

*Proof.* This follows by a simple application of Taylor's theorem as in the finite dimensional case. For Taylor's theorem for functions on Banach spaces, cf. (16). □

**Theorem 1.** *Suppose that* $\mathbf{F} \in \mathscr{C}^1(U; H)$*, that the derivative of* $\mathbf{F}$ *at* $\Phi \in U$ *is invertible, and* $\mathbf{F}(\Phi) = 0$*. There is a neighbourhood* $V \subset H$ *of* $0$ *small enough such that*

$$\mathbf{F}(\mathcal{N}) \longrightarrow 0 \implies \mathcal{N} \longrightarrow \Phi .$$

*Proof.* By Lemma 1, we can choose neighbourhoods $\Phi \in U' \subset U$ and $0 \in V \subset H$ such that $\mathbf{F} : U' \to V$ is a diffeomorphism. Then, if $\mathcal{N} \in U'$ the continuity of $\mathbf{F}^{-1}$ implies

$$\mathbf{F}[\mathcal{N}] \longrightarrow 0 \implies \mathcal{N} \longrightarrow \Phi .$$

which is the assertion. $\qquad\square$

**Theorem 2.** *Under the same assumptions as above we*

$$\|\mathcal{N} - \Phi\| = \mathcal{O}\left(\|\mathbf{F}[\mathcal{N}]\|\right) .$$

*Proof.* Since by Lemma 1, $\mathbf{F}^{-1}$ is differentiable at $0$, it follows that it is locally Lipschitz continuous around $0$, implying that

$$\|\mathcal{N} - \Phi\| = \mathcal{O}\left(\|\mathbf{F}[\mathcal{N}]\|\right) .$$

$\qquad\square$

The constant of proportionality is approximately given by $\lambda_{\min}^{-1}$ where

$$\lambda_{\min} \coloneqq \inf_{\lambda \in \mathrm{Spec}(\mathrm{D}\mathbf{F}[\Phi])} |\lambda|$$

the eigenvalue with minimal absolute value in the spectrum of $\mathrm{D}\mathbf{F}[\Phi]$. The nPBE example from section 6 fits nicely into this paradigm as the map

$$\mathbf{F} : W^{2,\infty}(\mathbb{R}^3) \longrightarrow W^{0,\infty}(\mathbb{R}^3)$$
$$f \longmapsto -\Delta f + \sinh(f) + g$$

is continuous and continuously Fréchet differentiable with Fréchet derivative

$$\mathrm{D}\mathbf{F}(f) = -\Delta + \cosh(f)$$

which is everywhere continuously invertible as a linear map. The constant of proportionality is approximately 1.

Theorems 1 and 2 guarantee that there exist adequate models for $\Phi$, if the loss is going down *sufficiently enough*. However, we still have no clue about how fast or slow this convergence is going to be and how to convert these existence results into actual estimates on $\Phi_{\epsilon_1}$ (and higher order $\Phi_{\epsilon_i}$). We develop those results below, with minimal additional assumptions.

### B.2  GRADIENT DESCENT IN HILBERT SPACES

We assume that $G$ is a Hilbert space, i.e. in addition to being a Banach space, it is also equipped with an inner product $\langle \cdot, \cdot \rangle$ such that $\langle v, v \rangle = \|v\|^2$ for all $v \in G$. Assume that we are given a loss function $\mathcal{L} : G \supset U \to \mathbb{R}$ that is an element of $\mathscr{C}^1(U)$. We denote by $\nabla\mathcal{L}(\mathcal{N})$ the unique element of $G$, s.t. for all $\Psi \in G$

$$\mathrm{D}\mathcal{L}[\mathcal{N}](\Psi) = \langle \nabla\mathcal{L}(\mathcal{N}), \Psi \rangle .$$

If $\nabla\mathcal{L}$ is a locally Lipschitz continuous or equivalently if $\mathcal{L}$ is a locally $L$-smooth function, then the ODE

$$\dot{\mathcal{N}}(t) = -\nabla\mathcal{L}(\mathcal{N}(t)) \tag{23}$$

has a unique local solution, cf. (17). By locally $L$-smooth, we mean that around a minimum $\Phi$, there exists $R > 0$ such that for all $\mathcal{N}, \mathcal{N}' \in B_R(\Phi)$

$$\left\|\nabla\mathcal{L}(\mathcal{N}) - \nabla\mathcal{L}(\mathcal{N}')\right\| \le L\|\mathcal{N} - \mathcal{N}'\| .$$

Furthermore, for $\mu > 0$, we call the function $\mathcal{L}$ locally $\mu$-strongly convex around a minimum $\Phi$, if there exists $R > 0$ such that for all $\mathcal{N}, \mathcal{N}' \in B_R(\Phi)$

$$\mathcal{L}(\mathcal{N}') \ge \mathcal{L}(\mathcal{N}) + \langle \nabla\mathcal{L}(\mathcal{N}), \mathcal{N}' - \mathcal{N} \rangle + \frac{\mu}{2}\|\mathcal{N}' - \mathcal{N}\|^2 .$$

We need one more lemma to have all the tools we will use to get Theorem 3.

**Lemma 3.** *Let $\mathcal{L} \in \mathscr{C}^1(U)$ be a locally $L$-smooth, locally $\mu$-strongly convex function around a minimum $\Phi$, s.t. $\mathcal{L} \geq 0$ and $\mathcal{L}(\Phi) = 0$.*

*Then for any initial condition $\mathcal{N} \in B_R(\Phi)$, a ball where $\mathcal{L}$ is both $L$-smooth and $\mu$-strongly convex, the Gradient Descent equation 23 converges exponentially at rate with rate $\frac{\mu}{2}$ towards $\Phi$, i.e.*

$$\|\mathcal{N}(t) - \Phi\| \leq e^{-\frac{\mu}{2}t}\|\mathcal{N}(0) - \Phi\| .$$

*Proof.* From the strong $\mu$-convexity it follows that

$$\frac{1}{2}\frac{d}{dt}\|\mathcal{N}(t) - \Phi\|^2 = -\langle\mathcal{N}(t) - \Phi, \nabla\mathcal{L}(\mathcal{N}(t))\rangle \leq -\underbrace{\mathcal{L}(\mathcal{N}(t))}_{\geq 0} - \frac{\mu}{2}\|y - x\|^2 \leq -\frac{\mu}{2}\|y - x\|^2 .$$

Solving this differential inequality, we find

$$\|\mathcal{N}(t) - \Phi\| \leq e^{-\frac{\mu}{2}t}\|\mathcal{N}(0) - \Phi\| .$$

$\square$

Lemma 3 guarantees the existence of an exponentially convergent regime of optimization, as long as we assume that $D\mathbf{F}$ exists and $\mathcal{L}$ is $\mu$-strongly convex. However, it is a non-constructive statement: we have no information on what behavior should be expected from $\mu$. We can make it constructive with an additional assumption on the existence of $D^2\mathbf{F}$.

### B.3    CONSTRUCTIVELY ESTIMATING THE RATE OF CONVERGENCE

Assume that $\mathbf{F}$ is in $\mathscr{C}^2(U; H)$. Furthermore, we require that $H$ is also a Hilbert space. From the definition of Fréchet differentiability we can write $\mathbf{F}$ as

$$\mathbf{F}[\mathcal{N}] = D\mathbf{F}[\Phi](\mathcal{N} - \Phi) + D^2\mathbf{F}[\Phi](\mathcal{N} - \Phi, \mathcal{N} - \Phi) + R_1(\mathcal{N}) ,$$

where $\|R_1(\mathcal{N})\| = o(\|\mathcal{N} - \Phi\|^2)$. Then our loss function $\mathcal{L}(\mathcal{N}) := \langle\mathbf{F}[\mathcal{N}], \mathbf{F}[\mathcal{N}]\rangle$ is strictly convex in a neighbourhood of $\Phi$ as

$$D^2\mathcal{L}[\Phi](\mathcal{N} - \Phi, \mathcal{N} - \Phi) = \langle D\mathbf{F}[\Phi](\mathcal{N} - \Phi), D\mathbf{F}[\Phi](\mathcal{N} - \Phi)\rangle$$
$$= \langle\mathcal{N} - \Phi, (D\mathbf{F}[\Phi])^\dagger D\mathbf{F}[\Phi](\mathcal{N} - \Phi)\rangle \geq \sigma_{\min}\|\mathcal{N} - \Phi\|^2$$

where we used that $\mathcal{L}(\Phi) = 0$ and defined

$$\sigma_{\min} := \sigma_{\min}(\Phi) := \inf_{\mathcal{N} \in G\setminus\{0\}} \frac{\|D\mathbf{F}[\Phi]\mathcal{N}\|^2}{\|\mathcal{N}\|^2} = \inf \operatorname{Spec}\left((D\mathbf{F}[\Phi])^\dagger D\mathbf{F}[\Phi]\right) > 0 ,$$

as $D\mathbf{F}[\Phi]$ is invertible. Furthermore, our loss function can be written as

$$\mathcal{L}(\mathcal{N}) = \langle D\mathbf{F}[\Phi](\mathcal{N} - \Phi), D\mathbf{F}[\Phi](\mathcal{N} - \Phi)\rangle + R_2(\mathcal{N}) =$$
$$= \langle\mathcal{N} - \Phi, (D\mathbf{F}[\Phi])^\dagger D\mathbf{F}[\Phi](\mathcal{N} - \Phi)\rangle + R_2(\mathcal{N})$$

where $|R_2(\mathcal{N})| = o(\|\mathcal{N}-\Phi\|^3)$ and $R_2$ is also a $\mathscr{C}^2$-function, and its second derivative is $o(\|\mathcal{N}-\Phi\|)$. Thus, for every $\epsilon > 0$ there exists $R > 0$ such that for all $\mathcal{N}$ in the ball of radius $R$ around $\Phi$

$$\frac{\|D^2R_2[\mathcal{N}]\|}{\min\{1, \|\mathcal{N} - \Phi\|\}} \leq \epsilon\sigma_{\min}.$$

Then, we have that for all $\mathcal{N} \in B_R(\Phi)$

$$\sigma_{\min}(\mathcal{N}) \geq (1 - \epsilon\min\{1, \|\mathcal{N} - \Phi\|\})\sigma_{\min} \geq (1 - \epsilon)\sigma_{\min} .$$

It follows that $\mathcal{L}$ is $\mu$-strongly convex in the ball of radius $R$ around $\Phi$ with constant $\mu = (1-\epsilon)\sigma_{\min}$, and thus the gradient descent flow $\mathcal{N}(t)$ converges exponentially at rate $\frac{(1-\epsilon)\sigma_{\min}}{2}$.

We have thus proven Theorem 3. Theorem 4 follows immediately from the results in appendix B.

## C  STRICT UPPER BOUNDS ON $\|\Phi_{\epsilon_1}\|$

We will begin our attempt to ascertain bounds on $\|\Phi_{\epsilon_1}\|$ with classical Hamiltonian systems, to exemplify the intuition and the technique involved in the generic bound we wish to develop. Once that is achieved, we will show how a generalization follows in a straightforward manner.

### C.1  HAMILTONIAN SYSTEMS

To start, let us remember $\Phi_{\epsilon_1} := \mathcal{N} - \Phi = (N_1 - \phi_1, \ldots, N_D - \phi_D)$ for a $D$-dimensional dynamical system to be solved on a domain of interest $[0, T]$. Note that $D$ is necessarily even for a Hamiltonian dynamical system. Let us construct a worst case scenario for the norm of $\Phi_{\epsilon_1}$. We write the equation in terms of the Hamiltonian formulation:

$$\frac{d\Phi}{dt} = J\nabla\mathcal{H}(\Phi) \tag{24}$$

where $\mathcal{H}$ is the appropriate Hamiltonian, and $J$ is the symplectic matrix

$$J = \begin{pmatrix} 0 & -I_{D/2} \\ I_{D/2} & 0 \end{pmatrix} \tag{25}$$

and $I_{D/2}$ is the identity matrix. The NN DE solver is trained using $\mathbf{F}[\mathcal{N}]$ given by

$$\mathbf{F}[\mathcal{N}] = J\nabla\mathcal{H}(\mathcal{N}) - \frac{d\mathcal{N}}{dt} \tag{26}$$

The equation above represents $D$ separate differential equations in a vector form. Since $\mathbf{F}[\Phi](t) = 0$ for all $t$ we can write $\mathbf{F}[\mathcal{N}]$ as follows, suppressing the time dependence

$$\mathbf{F}[\mathcal{N}] = -\int_0^1 \mathrm{D}\mathbf{F}[\mathcal{N}_s]\Phi_{\epsilon_1}ds = \dot{\Phi}_{\epsilon_1} - \left[\int_0^1 J\left(\mathrm{D}^2\mathcal{H}\right)(\mathcal{N}_s)ds\right] \cdot \Phi_{\epsilon_1} =: \dot{\Phi}_{\epsilon_1} - R\Phi_{\epsilon_1} \tag{27}$$

where $\mathcal{N}_s := \Phi - s\Phi_{\epsilon_1}$, and $\mathrm{D}^2\mathcal{H}(\Phi)$ is the Hessian matrix of $\mathcal{H}$ with components $\partial_i\partial_j\mathcal{H}(\mathcal{N})$. In order to be able to extract a meaningful error bound from Eq. equation 27 we need to make some structural assumption: We assume that at a time $t_{\Phi_{\epsilon_1}} \in [0, T]$ at which $\|\Phi_{\epsilon_1}\|$ obtains its maximum

$$\left[R(t_{\Phi_{\epsilon_1}})\Phi_{\epsilon_1}(t_{\Phi_{\epsilon_1}})\right] \cdot \dot{\Phi}_{\epsilon_1}(t_{\Phi_{\epsilon_1}}) = 0$$

Note that the assumption of focusing all error in only one component of $\Phi$, as done in (10), already implies the above, but not vice versa. As such, we are using a much weaker assumption than (10). [3] Then, for time $t_{\Phi_{\epsilon_1}} \in [0, T]$, we have

$$\|R\Phi_{\epsilon_1}\|_2^2 \le \|R\Phi_{\epsilon_1}\|_2^2 + \|\dot{\Phi}_{\epsilon_1}\|_2^2 = \|\mathbf{F}[\mathcal{N}]\|_2^2 \tag{28}$$

We next need an estimate on the matrix norm of the inverse of $R$ for every $t \in [0, T]$. We can ignore $J$ as it is an orthogonal matrix and only consider

$$\int_0^1 \left(\mathrm{D}^2\mathcal{H}\right)(\mathcal{N}_s)ds \ .$$

We can do this if we know that $\mathrm{D}^2\mathcal{H}(x)$ is a strictly positive or strictly negative definite matrix for all $x \in \mathcal{D} \subset \mathbb{R}^D$ in our domain of interest, which is a standard assumption for the motion to be non-degenerate. W.l.o.g., assuming that $\mathrm{D}^2\mathcal{H}$ is positive definite, and setting

$$H_{\min} := \min_{x \in \mathcal{D}} \|\mathrm{D}^2\mathcal{H}(x)^{-1}\|^{-1} = \min_{x \in \mathcal{D}} \lambda_{\min}(\mathrm{D}^2\mathcal{H}(x))$$

where $\lambda_{\min}$ is the smallest eigenvalue of the matrix $\mathrm{D}^2\mathcal{H}(x)$, we have for all $t \in [0, T]$

$$\|R^{-1}\| \le H_{\min}^{-1} \ .$$

---

[3]Unfortunately, we need this assumption to separate the two terms appearing on the right hand side of Eq. equation 27.

Finally, letting

$$F_{\max} := \max_{t \in [0,T]} \|\mathbf{F}[\mathcal{N}](t)\|_2$$

we reach the an estimate on $\|\Phi_{\epsilon_1}\|$ as:

$$\|\Phi_{\epsilon_1}\| \le \frac{F_{\max}}{H_{\min}} \tag{29}$$

Additionally, if we assume that at the time $t_{\dot{\Phi}_{\epsilon_1}}$ at which $|\dot{\Phi}_{\epsilon_1}|$ reaches its maxima,

$$\left[R(t_{\dot{\Phi}_{\epsilon_1}})\Phi_{\epsilon_1}(t_{\dot{\Phi}_{\epsilon_1}})\right] \cdot \dot{\Phi}_{\epsilon_1}(t_{\dot{\Phi}_{\epsilon_1}}) = 0$$

we have the following bound on $\dot{\Phi}_{\epsilon_1}$ as well

$$\|\dot{\Phi}_{\epsilon_1}\| \le F_{\max} \tag{30}$$

## C.2 Bounds on $\|\Phi_{\epsilon_1}\|$ in more general settings

Eq. 27 generalizes beyond Hamiltonian systems in a straightforward manner for $\mathbf{F}[\ ] \equiv \mathbf{L}[\ ] + \mathbf{N}[\ ] + \mathbf{C}$ as:

$$\mathbf{F}[\mathcal{N}] = -\int_0^1 \mathrm{D}\mathbf{F}[\mathcal{N}_s]\Phi_{\epsilon_1} ds = \left[\int_0^1 \mathrm{D}\mathbf{N}[\mathcal{N}_s] ds\right] \cdot \Phi_{\epsilon_1} - \mathbf{L}[\Phi_{\epsilon_1}] =: -R\Phi_{\epsilon_1} - \mathbf{L}[\Phi_{\epsilon_1}] \tag{31}$$

The generalized version of the $R\Phi_{\epsilon_1} \cdot \dot{\Phi}_\epsilon = 0$ assumption is that at the $\mathbf{x}_{\Phi_{\epsilon_1}} \in \mathcal{D}$ that $\|\Phi_{\epsilon_1}\|$ takes its maxima in, we have:

$$\left[R(\mathbf{x}_{\Phi_{\epsilon_1}})\Phi_{\epsilon_1}(\mathbf{x}_{\Phi_{\epsilon_1}})\right] \cdot \mathbf{L}[\Phi_{\epsilon_1}](\mathbf{x}_{\Phi_{\epsilon_1}}) = 0 \qquad \mathbf{x}_{\Phi_{\epsilon_1}} \in \mathcal{D}$$

The next assumption we generalize is the one made to have non-degeneracy in the solutions we were trying to model. As such, we assume that $\mathrm{D}\mathbf{N}$ is a positive definite (or negative definitive) operator. From then on, we define $H_{\min}$ in a similar manner. More precisely, we say:

$$\begin{aligned} H_{\min_1} &:= \inf_{s \in [0,1]} \inf \left\{\lambda \,\big|\, \lambda \in \mathrm{Spec}\left(|\mathrm{D}\mathbf{N}[\mathcal{N}_s]|\right)\right\} \\ H_{\min_2} &:= \inf_{s \in [0,1]} \inf \left\{\lambda \,\big|\, \lambda \in \mathrm{Spec}\left(|\mathbf{L}[\Phi_{\epsilon_1}]|\right)\right\} \end{aligned} \tag{32}$$

where

$$|\mathrm{D}\mathbf{N}[\mathcal{N}_s]| := \sqrt{(\mathrm{D}\mathbf{N}[\mathcal{N}_s])^\dagger \mathrm{D}\mathbf{N}[\mathcal{N}_s]}$$

Finally, let $H_{\min} = \max\{H_{\min_1}, H_{\min_2}\}$. With that in place, we obtain the following inequalities for $\|\Phi_{\epsilon_1}\|$:

$$\|\Phi_{\epsilon_1}\| \le \frac{F_{\max}}{H_{\min}} \tag{33}$$

The generalized version of the assumption on $\mathbf{x}_{\dot{\Phi}_\epsilon}$ where $|\dot{\Phi}_\epsilon|$ attains its maxima similarly gives:

$$\|\dot{\Phi}_\epsilon\| \le F_{\max} \tag{34}$$

The assumption that $\left[R(\mathbf{x}_{\Phi_{\epsilon_1}})\Phi_{\epsilon_1}(\mathbf{x}_{\Phi_{\epsilon_1}})\right] \cdot \mathbf{L}[\Phi_{\epsilon_1}](\mathbf{x}_{\Phi_{\epsilon_1}}) = 0$ is critical in allowing us strong estimates on $\|\Phi_{\epsilon_1}\|$. Superficially, it might seem like too strong of an assumption (and even somewhat of a non-sequitur).

However, note that $\mathbf{L}[\ ]$ in a DE is often the domain derivative term (we denote it as $\nabla_\mathbf{x}$ in this discussion, where $\mathbf{x} \in \mathcal{D}$ represents an arbitrary element in the domain. As such, $\nabla_\mathbf{x}$ are the spatial derivatives for spatial DEs, time derivatives for ODEs, a spatio-temporal one for space-time DEs, etc). When $\|\Phi_{\epsilon_1}\|$ achieves its maxima, $\Phi_{\epsilon_1} \cdot \nabla_\mathbf{x}(\Phi_{\epsilon_1}) = 0$. When $\mathbf{L} \equiv \nabla_\mathbf{x}$, we indeed have

$\Phi_{\epsilon_1} \cdot \mathbf{L}(\Phi_{\epsilon_1}) = 0$. As such, what we are really assuming is that the $R$ term effectively has $\Phi_{\epsilon_1}$ as its eigenvector on $\mathbf{x}_{\Phi_{\epsilon_1}}$ (since that means $R\Phi_{\epsilon_1} \cdot \nabla_{\mathbf{x}}(\Phi_{\epsilon_1})$ is 0). For example, such is the case for Hamiltonian systems without mixed position-momentum terms. What we are really constraining, is the behavior exhibited by the operator $\mathbf{N}$ in a system of interest.

If the problem of interest comes from a scientific domain with which the user has familiarity, they may supply a different sort of assumption based on those considerations. For example, we might be able to say:

$$\|R\Phi_{\epsilon_1} \cdot \nabla_{\mathbf{x}}\Phi_{\epsilon_1}\| < \frac{\epsilon}{2}$$

at the $\mathbf{x}_{\Phi_{\epsilon_1}}$ where $\|\Phi_{\epsilon_1}\|$ takes its maxima. The generalized bound simply changes to:

$$\|\Phi_{\epsilon_1}\| \leq \frac{F_{\max} + \epsilon}{H_{\min}} \tag{35}$$

# D   ADDITIONAL NUMERICAL EXPERIMENTS

As part of validating our claims, we also performed numerical experiments on several scientific ODEs and PDEs. We chose our experiments such that the set of examples showcases non-trivial spatial and temporal phenomena amongst the combined examples (chaos, high dimensional domains, etc).

Table 1 presents the median results from the collection of randomized experiments conducted for each system. All numerical experiments were done using a 2019 MacBook Pro with a 2.6 GHz 6-Core Intel Core i7 processor and 16 GBs of 2667 MHz DDR4 RAM.

We describe the chosen systems below:

## D.1   HENON HEILES

Henon Heiles is represented by the following ODE governing the dynamics of $\Phi \equiv \{x, y, p_x, p_y\}^{\dagger}$:

$$\dot{\Phi} = -J\nabla\mathcal{H}, \quad J = \begin{pmatrix} \mathbf{0} & \mathbf{I} \\ -\mathbf{I} & \mathbf{0} \end{pmatrix}, \quad \mathcal{H} \equiv \frac{x^2 + y^2 + p_x^2 + p_y^2}{2} + \frac{y(3x^2 - y^2)}{3} \tag{36}$$

where $\mathbf{I}$ is the $2 \times 2$ identity matrix. We picked $[0, 6\pi]$ as the time domain of interest. All $\Phi(0) \equiv \{x(0), y(0), p_x(0), p_y(0)\}$ were picked s.t. $\mathcal{H}(\mathbf{z}(0)) < \frac{1}{6}$, $x(0) \in [-\frac{\sqrt{3}}{2}, \frac{\sqrt{3}}{2}]$, and $y(0) \in [-0.5, 1 - \sqrt{3}|x|]$, which corresponds to the subset of the phase space that has bounded orbits.

We implement the NN DE solver prescribed in (10). The models each had 2 hidden layers, with 50 sine activation functions per layer. The base models were trained for 50000 iterations using ADAM, with an error correction made at 25000 iterations. We sampled 200 time points were iteration.

## D.2   NONLINEAR POISSON BOLTZMANN EQUATION

We have already given details of the equation in the main paper, alongside the associated choices of dimensionality and frequency parameter. Below, we plot what the $(\omega, d) = (5, 2)$ system should look like when modeled without (a) and with (b) error correction, alongside the true solution (c). To quantify the visual, we also quote the mean average value of $\|\mathcal{N}\|$, $\|\mathcal{N} + \mathcal{N}_{\epsilon_1}\|$, $\|\Phi\|$ respectively.

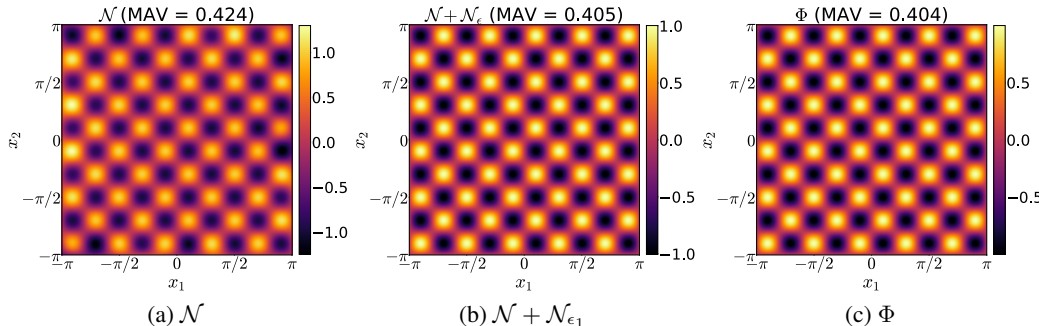

(a) $\mathcal{N}$           (b) $\mathcal{N} + \mathcal{N}_{\epsilon_1}$           (c) $\Phi$

Figure 3: Heatmaps of non-error corrected $\mathcal{N}$ & error corrected solver with $\mathcal{N}_{\epsilon_1}$ at 50% of the total training iterations (left & middle), and $\Phi$ (right) for the $(\omega, d) = (5, 2)$ setting.

$\mathcal{N}$ and $\mathcal{N}_{\epsilon_1}$ were fully connected Neural Networks, with 4 hidden layers, each with 50 sine activation functions. The base model was trained via ADAM for 400000 iterations, while error models were trained for a varying number of iterations, depending on when the error correction models were activated (see Fig. 2). We sampled 1024 points per iteration.

The $(\omega, d) = (1, 4)$ model was trained for 50000 iterations, with 8192 points sampled per iteration.

Both NN DE solvers are implemented using the methods prescribed in (2)

### D.3 NONLINEAR QUARTIC OSCILLATOR

The nonlinear quartic oscillator is represented by the following ODE governing the dynamics of $\Phi \equiv \{x, p_x\}$:

$$\dot{\Phi} = -J\nabla\mathcal{H}, \quad J = \begin{pmatrix} 0 & 1 \\ -1 & 0 \end{pmatrix}, \quad \mathcal{H} \equiv \frac{x^2 + p_x^2}{2} + \frac{x^4}{4} \tag{37}$$

We sample $\Phi(0)$ such that $\{x(0), p_x(0)\} \in [-1, 1] \times [-1, 1]$ each time.

We implement the NN DE solver prescribed in (10). The models had 2 hidden layers each, with 50 sine activation functions per layer. The base models were trained for 50000 iterations using ADAM, with an error correction made at 25000 iterations. We sampled 200 time points were iteration.

The associated codebase allows the user to add other systems as per their choice.

