# OpenReview forum: "Neural Network Differential Equation Solvers allow unsupervised error estimation and correction"
_ICLR.cc/2023/Conference — Submitted to ICLR 2023_

### Official Review · Reviewer_25yz · 2022-10-20

**Confidence:** 3
**Correctness:** 3
**Technical Novelty And Significance:** 3
**Empirical Novelty And Significance:** 3
**Recommendation:** 6

**Clarity, Quality, Novelty And Reproducibility:**

The paper is very well-written, and addresses one of the most fundamental questions in the field. The writing is very lucid.

Sufficient details are provided to reproduce the experiments.


**Strength And Weaknesses:**

Overall, the paper is very well-written. Despite having a lot of dense theorems, overall the explanation is clear, and it is very well-connected to practical implications.

I have a few questions:
- Can you bound how much the solution error goes down at each stage of estimating the error of the previous network? How do we know we won’t get some degenerate behaviour, e.g. where the networks in the chain just alternate between overcorrecting each other.
- Are there ablations to other simpler methods for training NN solvers? For example, your approach requires K models to perform K - 1 steps of error correction. If we just take one model that is (roughly) K times as big and train it once, how does the solution error compare?


**Summary Of The Paper:**

This is primarily a theoretical paper addressing the problem of solving differential equations with neural networks. Existing approaches do not provide any guarantees about their error relative to the true solution. Specifically, optimising the loss function doesn’t bound the error of the solution.

Section 2 of the paper sets up a formalism and notation used throughout the rest of the paper.

Section 3 describes some basic results about bounding the error of a NN solution. Specifically, some asymptotic results are given about the loss going down with the solution error of the neural net, and a bound is given for the solution error dependent on Spec of the non-linear portion of the DE.

Section 4 describes the main contribution of the paper. It describes a way to give more useful bounds on the solution error, without knowing the solution already. Essentially, they describe a simple procedure to define a new DE whose unique solution is the error of the NN solver on the original problem. We can extend this to create a chain of networks, each estimating the error of the solution defined so far (i.e. the sum of the original solution, and subsequent error estimates).

Section 5 describes some potential extensions of the work.

Section 6 shows numerical experiments.

**Summary Of The Review:**

Overall, the paper is very well-written. Despite having a lot of dense theorems, overall the explanation is clear, and it is very well-connected to practical implications.

I have a few concerns regarding some additional ablations (see Strengths and Weaknesses).

I have not checked the proofs of any of the theorems, and I do not have an extensive theory background, hence the low confidence in my score.

---

> ### Author Response · Authors · 2022-11-19
> **Response to Reviewer 25yz**
>
> We thank the reviewer for their very positive comments, and doing their best to evaluate the work, despite the fact that it falls outside their background. We address the raised concerns below:
>
> 1. "Can you bound how much the solution error goes down at each stage of estimating the error of the previous network? How do we know we won’t get some degenerate behaviour, e.g. where the networks in the chain just alternate between over-correcting each other."
>
> Theorem 4 provides the result for generic order error modelling bounds, alongside the guarantees that gradient descent will only take us closer and closer. Practically speaking, since we perform error correction on the best model until each order, we are guaranteed to at least the performance we have already achieved at every step: only a monotonic decrease in error is possible.
>
> 2. "Are there ablations to other simpler methods for training NN solvers? For example, your approach requires K models to perform K - 1 steps of error correction. If we just take one model that is (roughly) K times as big and train it once, how does the solution error compare?"
>
> We have added ablation experiments (for both size and optimization time) as per your advice and find that error correction significantly outperforms them. The ROI metric in Table 1 explicitly showcases the value added by error correction vs the ablation choices.

---

### Official Review · Reviewer_6cUn · 2022-10-25

**Confidence:** 1
**Correctness:** 4
**Technical Novelty And Significance:** 4
**Empirical Novelty And Significance:** 3
**Recommendation:** 8

**Clarity, Quality, Novelty And Reproducibility:**

The paper is well written. However, this work could benefit from a better contextualization of the results relative to a real-world problem.. The results seem to be novel, high quality, and reproducible.

**Strength And Weaknesses:**

STRENGTHS:
* The algorithmic ideas proposed in the work are novel, technically sound, and non-trivial. The depth of the technical results and their derivations is significant.
* The experiments show promising results against existing baselines.

WEAKNESSES:
* This work could benefit from a better contextualization of the results relative to a real-world problem.
* The experiments section could benefit from additional analysis on downstream applications of the proposed solvers.

**Summary Of The Paper:**

The authors propose techniques for building more efficient neural network differential equation solvers based on a novel analysis of the optimization objective used by existing solvers. The authors study and derive new bounds, equations, and convergence rate for existing solvers and use these to propose a new algorithm. They provide numerical experiments that show improvements over existing baselines.

**Summary Of The Review:**

I can imagine this paper being a useful addition to this line of literature, but this is outside my scope of expertise, and I would mostly defer to the other reviewers.

---

> ### Author Response · Authors · 2022-11-19
> **Response to Reviewer 6cUn**
>
> We thank the referee for their very positive comments, and doing their best to evaluate the work, despite the fact that it is outside their field.
>
> We agree that real world examples would be an excellent addition to the work. However, ICLR constraints on length mean that we see no way to incorporate such items in this work, without dropping other essential items. As such, we apologize for being unable to address their concern.

---

> > ### Comment · Reviewer_6cUn · 2022-12-08
> > **Thanks**
> >
> > I acknowledge reading the response.

---

### Official Review · Reviewer_3Hkr · 2022-11-01

**Confidence:** 5
**Correctness:** 3
**Technical Novelty And Significance:** 3
**Empirical Novelty And Significance:** Not applicable
**Recommendation:** 3

**Clarity, Quality, Novelty And Reproducibility:**

The presentation is acceptable overall. However, there are some parts to be enhanced, e.g., a weird line between Eq 2 and 3. Their findings and algorithms are novel.

**Strength And Weaknesses:**

- S1) As I know, this paper is the first to measure the error level of NNDE solvers in an unsupervised manner.
- S2) Their findings are novel, and their error estimation method is easy to follow and seems effective.

- W1) Experimental results are weak. They need to test with PINN models.
- W2) Some PINNs use additional loss functions other than F. In such a case, I am unsure if the proposed method can still be applied.
- W3) Do you consider boundary conditions?
- W4) The correction method is iterative, which requires large resources.

**Summary Of The Paper:**

NNDE solvers are popular these days, e.g., PINN, etc. This paper presents a method to estimate the error of those NNDE solvers and an iterative method to improve their accuracy. The authors introduce two fundamental theorems, from which their algorithms are designed. They also present some experimental results to show the correctness of their findings and algorithms.

**Summary Of The Review:**

I think this paper has non-trivial value in the PINN community. I like their overall idea to estimate the error level and iteratively correct in an unsupervised manner. However, the biggest problem is their experiments. To be accepted, I think they need to test with many PINN variants. Ever since the first introduction of PINN, there have been many other variants. The authors should show the efficacy of their method to them. If not, their nice findings cannot be appreciated. Please answer the weak points above to help my understanding. During the rebuttal period, I recommend that the authors work hard to test with real PINN models. I will reconsider my score after the rebuttal since this paper has non-trivial value in my opinions with one critical problem in their experiments.

---

> ### Author Response · Authors · 2022-11-05
> **Request for clarification from Reviewer 3Hkr**
>
> We thank you for the time and effort you have put into reviewing our work. We intend to make an in-depth response to your concerns promptly. However, we would be grateful if you could clarify one critical point for us. When you recommend that we do more experiments, do you mean to say that we should
>
> 1) Showcase the validity of methods on more Differential Equations.
> OR
> 2) Showcase the utility of our results by showing that inequality estimates and/or error models can be generated for codebases (PINNs) that other people have written.
>
> If you mean the latter, we would be grateful if you could let us know which codebases (PINNs) you have in mind as being particularly convincing.
>
> Our results came after an attempt at building an effective theory of NN DE solvers. As long the generic assumptions of our theory are satisfied (which they should be in almost all cases), the results should be valid.

---

> ### Author Response · Authors · 2022-11-19
> **Response to Reviewer 3Hkr**
>
> Thank you for your efforts in reviewing our work. We respond to each criticism, clarification, or modification you raised below:
>
> W1) Experimental results are weak. They need to test with PINN models.
>
> We have used different kinds of DEs and adapted two kinds of PINNs to stress-test these ideas. Since our assumptions are codebase invariant, we expected success in each of them and the results bear that out. In those settings, we have now also added ablation studies to rule out other possible candidates for the success of error correction as a method.
>
> However, we are also taking your recent advice and are working to adapt more NN DE solver methods (and bench-marking results against the ones reported in their respective papers). We will update the results as they come in and thank you for your patience in this regard.
>
> W2) Some PINNs use additional loss functions other than F. In such a case, I am unsure if the proposed method can still be applied.
>
> Our theorems have now been generalized to every kind of loss function design that relies on a mapping ${\bf F}$ that satisfies $F(N) = 0, \implies N = \Phi$ and satisfies the assumptions on $\mathrm{D}{\bf F}$ and $\mathrm{D}{\bf F}^{-1}$ (for example the loss could be based not on the equation operator as used by most codebases, but some complicated function of the equation operator. It could even be based on some sort of data-based consideration, as long as the other assumptions are obeyed).
>
> W3) Do you consider boundary conditions?
>
> We consider both approaches: boundary conditions being enforced exactly (subsection 2.3) and the boundary conditions enforced using some loss term (usually a simple regression function). Indeed, we generalize the notion to \textit{constraint conditions}, where any known information might be enforced.
>
> W4) The correction method is iterative, which requires large resources.
> Error correction, at least as prescribed in practice by us, requires a theoretical maximum of 2X increase in resources relative to the standard method. Empirically, we have found that it only leads to 20-30\% increase. Since the returns are measured in orders of magnitude, we believe error correction more than pays for itself. We have added an ROI term in Table 1 to quantify the value added by error correction in each system.

---

> ### Author Response · Authors · 2022-12-12
> **Final Response to Reviewer 3Hkr**
>
> We thank the reviewer for raising several interesting points that directly led us to a better version of the work (and hope we were able to satisfactorily resolve all bar the one discussed below). Unfortunately, due to personal/familial circumstances affecting multiple authors, we have been unable to conduct any new numerical experiments since our update on Nov 18. Indeed, we were not even able to get anything started on the suggested XPINNs or CPINNs. As such, we only report results on two types of PINN variants (HNNs with exact ICs and standard PINNs). We hope the discussions around every other point coupled with the existing numerical evidence leads to sufficient trust on the reviewer's behalf on the theory we have built for NN DE solvers. Irrespective of the decision, we thank the reviewer for their efforts on this work.

---

### Official Review · Reviewer_cgGX · 2022-11-07

**Confidence:** 4
**Correctness:** 2
**Technical Novelty And Significance:** 3
**Empirical Novelty And Significance:** 3
**Recommendation:** 3

**Clarity, Quality, Novelty And Reproducibility:**

I found in the beginning of the paper somewhat confusing. In particular, the discussions in section 2 and 3 seem largely immaterial to the proposed method. I found them to be rather confusing than providing helpful motivation.

This, as well as spurious claims and vague language such as "However, this is still substantially better than the situation that existed before: even an ambiguous estimate of $\Phi_{\epsilon_1}$ has clear practical benefits".

The proposed method appears to be novel and surprisingly effective.

**Strength And Weaknesses:**

### Strengths

- To the best of my knowledge, the methodology proposed by the authors is novel.
- The empirical performance of the proposed method is very promising.

### Weaknesses

- It seems unclear that the theory presented in section 3 is applicable to any PDE
- The claim of an improved error estimation does not seem to hold up.

**Summary Of The Paper:**

This article proposes a novel method for training neural networks to solve partial differential equations. It begins by pointing out the necessity to infer from the vanishing PDE residual the vanishing of the error of the solution of the PDE.

It begins by providing theorems that, under certain conditions, allow bounding the error as a function of the residual.
It then proceeds to attempt to obtain better estimates for the error by plugging it into the PDE, and thus obtaining a PDE (closely related to the original PDE) that has to be satisfied by the mismatch between the true solution and the PDE.

By introducing a *second* neural network that is trained to solve for this mismatch and adding its results to the first neural network, the authors obtain remarkable improvements in accuracy.

**Summary Of The Review:**

I am very much on the fence about this work. On the one hand, the proposed method is showing surprisingly clear gains, despite being very simple. On the other hand, the present work contains too many spurious claims or handwavy claims and too little insight into why the method works. I therefore believe that it is not ready for publication in the present state.

More concretely, my main points of criticism are summarized below:

1. Not a single example is given for the application of Theorem 1. In particular, I was not able to discern what the norm is denoted by $\|\cdot\|$ throughout the paper. When using the norms from elsewhere in the paper (p-norms for the residuals, L^2 error for the solution error in the final results), not even a linear differential operator will satisfy the assumptions of the theorem.

2. The paper leads with the promise of improving the error estimation in NN-based PDE solvers. However, as the authors themselves remark, this problem is merely hidden in the "error estimate of the error estimate". I was not able to observe any argument as to why estimating the error of the error should be any easier than that of the original error. To this end, the authors seem to only produce the unsubstantiated claim that "... this is still substantially better than the situation that existed before: even an ambiguous estimate of $\Phi_{\epsilon_1}$ has clear practical benefits".

3. It is quite surprising that the proposed method achieves the improvements reported by the authors. It would therefore be very valuable to provide additional insights. For instance, it would be helpful to provide the derivation of the PDEs for the error corrections in the case of one particular PDE. Also, additional ablation studies would be welcomed, for instance understanding how much of the improvement simply arises from the architectural change of representing the solution as the sum of the outputs of two neural networks.

Overall, I believe that the work shows promising results but is not yet ready for publication in ICLR.

==============================================================
The discussion with the authors has further confirmed my concerns about the presentation of the paper. I therefore changed my recommendation to 3 (reject)

---

> ### Author Response · Authors · 2022-11-19
> **Response to Reviewer cgGX**
>
> Thank you for your efforts in reviewing our work. We respond to each criticism, clarification, or modification you raised below:
>
> 1. "It seems unclear that the theory presented in section 3 is applicable to any PDE"
>
> Note that our assumptions only concern $\bf F$ and $\mathrm{D}{\bf F}$: whether ${\bf F}$ comes from an ODE or PDE is immaterial. However, we have added explicit examples of the DEs (including nonlinear PDEs) that are rigorously within the remit of this work. The nPBE, to which an entire section is devoted as a means of exemplification, is one such nonlinear PDE.
>
> 2. "the discussions in section 2 and 3 seem largely immaterial to the proposed method"
>
> We apologize for the confusion caused by Section 2/3 and have done our best to better contextualize them (including the incorporation of the comment on giving examples). However, please note that the work is about both error analysis and correction. Before proposing a method for constructing unsupervised error models, we had to first prove that such a thing is mathematically possible (and ascertain the assumptions under which it would be reliably so). As such, Sections 2 and 3 are as important, if not more so, than the algorithm and numerical experiments. Indeed, the proposed methods rely on the principle that the equation residual ${\bf F}$ is a strong signal for the error in the model, even though at first glance the two are only implicitly related. Sections 2 and 3 make that intuition precise and tell us how to search for the model error without knowing the solution.
>
> 3."On the other hand, the present work contains too many spurious claims or handwavy claims and too little insight into why the method works."
>
> We have done our best to update and clarify the discussion within the text and relate it back to the proven results. We now clearly emphasize why error analysis+correction is a strictly more principled approach than the usual NN DE solver methods due to the generalized results of Theorem 4 (part of text originally as well, but perhaps not properly referenced).
>
> 4. "Not a single example is given for the application of Theorem 1. When using the norms from elsewhere in the paper (p-norms for the residuals, L^2 error for the solution error in the final results), not even a linear differential operator will satisfy the assumptions of the theorem."
>
> The section and numerical experiments on the nPBE were included as an explicit example of the relevance of the results. We have reworked the discussion to emphasize the fact clearly.
>
> The reason that the norm $\| \cdot \|$ is not specified in most instances is that we are working over generic Banach spaces and maps between them. Thus, the norm $\| \cdot \|$ denotes the norm of the Banach space to which the corresponding element belongs. When giving specific examples we specify the Banach spaces (and thus their norms).
>
> Furthermore, a linear differential operator of order k is a bounded (and thus smooth) operator from the Sobolev space $W^{p,k} \to W^{p,0} = L^p$. What you seem to be referring to is that a linear differential operator (and thus any derived non-linear PDE) cannot be a bounded operator from a space to itself, which is not a setting we are working in though (cf. end of appendix B.1 for more discussion on the nPBE example).
>
> 5. "I was not able to observe any argument as to why estimating the error of the error should be any easier than that of the original error".
>
> Theorem 4 explicitly proves why the shift in ambiguity is not a problem for systems where original assumptions hold. For systems they don't hold in, the issue is only as bad as the one in the usual method: as such, any improvement error correction can provide necessarily leads to a better outcome.
>
> The error models provide fine scale adjustments. Further optimization of converged base models is hindered in how much it can search for better models, because of already established performance. However, the parameters of the error model have no interaction with those of the base model and thus have more flexibility as they search for the fine adjustments needed. Real NN models can't have their exponential convergence phase last forever (see intro to appendix B). Further, the capacity of a slightly larger model looking for the same $\Phi$ is not going to be orders of magnitude more. However, error correction sidesteps this because the new parameters are looking for a different solution entirely and have an untapped exponential convergence phase. Lastly, they have ${\bf F}[\mathcal{N}]$ to refine that search and ${\bf F}_1 [\mathcal{N}_e]$ to reliably track it.
>
>
> 6. "For instance, .. sum of the outputs of two neural networks."
>
> One example has already been provided via the nPBE for getting ${\bf F}_1$: general procedure directly follows from footnote 1.
>
> Two kinds of ablation experiments (the word is used in a loose sense) are now in place and the results verify the expected utility of error correction.

---

> > ### Comment · Reviewer_cgGX · 2022-11-30
> > **Thank you for your response.**
> >
> > Thank you for your responses. I appreciate your efforts to address my concerns. However, I continue to have doubts about the paper in its present form. My most urgent concern is the following:
> >
> > On page 13, you claim that the linear map $-\Delta + \cosh(f)$ is boundedly invertible (as a map from $W^{2, \infty}$ to $W^{0, \infty}$). I am not aware of higher elliptic regularity theory in sobolev spaces with exponent $p = \infty$. Could the authors please provide a reference for this claim?

---

> > > ### Comment · Reviewer_cgGX · 2022-11-30
> > > **I am increasingly concerned**
> > >
> > > After a look through the literature, the claimed regularity result is certainly false in the case of $-\Delta$ (without the $\cosh$ term). I am therefore highly suspicious that the result is true as claimed for arbitrary $f \in L^{\infty}$.
> > > If the result claimed by the authors (without proof or a reference) turns out to be false, this casts doubt on the thoroughness of the work and thus constitutes a major red flag. I have lowered my score to strong reject until receiving a compelling answer by the authors.

---

> > > > ### Author Response · Authors · 2022-12-02
> > > > **Existence of the bounded inverse of DF for the nPBE**
> > > >
> > > > Thanks for outlining your major concern. We omitted a full proof on the bounded inverse of the nPB equation, because we believed it was tangential to this work. Within the context of this paper, it was too fringe+involved a proof to merit introducing the needed technical machinery, since it would have no connection to the rest of the paper. Simultaneously, it is also too straightforward to have been a generic result in the existing functional analysis literature: to the best of our knowledge, it is not explicitly shown to be true in any previous work. While we believe that decision was justified in terms of presenting the paper, we will be very happy to add the rigorous proof in the comments below to our manuscript, if the reviewers decide that it is wise.
> > > >
> > > > We also thank you for raising the issue, because it allowed us to identify a slight typo in the appendix and possibly the source of your dominant concern. The bounded inverse result for our setting/paradigm is for
> > > > \begin{align*}
> > > >     {\bf F} : H^{2}(\mathbb{R}^3) & \longrightarrow L^{2}(\mathbb{R}^3) , \qquad \qquad f \longmapsto -\Delta f + \sinh(f) + g  .
> > > > \end{align*}
> > > > Unfortunately, the typo must have escaped us during version control and caused a confusion regarding the relevant spaces. We have now corrected the typo.
> > > >
> > > > Note that the typo/correction does not have any bearing on any components of the rest of the work, since here we are only verifying that the nPBE satisfies the assumption of the theorem. The corrected sentence (alongside the proof) show that the nPBE is indeed a good example to use, but do not cause a change anywhere else.
> > > >
> > > > We believe the proof below should fully resolve your concerns, but we remain eager to provide more details or expand on anything else that you believe needs addressing.

---

> > > > > ### Author Response · Authors · 2022-12-02
> > > > > **Proof**
> > > > >
> > > > > $\bf F$ is well-defined and continuous as long as $g \\in L\^2(\\mathbb{R}\^3)$, since $f \\in H\^2(\\mathbb{R}\^3)$ implies that $\\sinh(f) \\in L\^2(\\mathbb{R}\^3)$. To see the latter, note that for $x \\in [-1,1]$, $|\\sinh(x)| \\leq \\cosh(1)|x|$. On the other hand, by the Sobolev inequality on $\\mathbb{R}\^n$, cf. [1, Theorem 4.12], there exists a constant $C > 0$, s.t. $||f||\_{L\^{\\infty}} \\leq C ||f||\_{H\^2}$.
> > > > >
> > > > > Thus, $||\\sinh(f)||\_{L\^\\infty} \\leq \\sinh(C ||f||\_{H\^2})$, by the monotonicity of $\\sinh$. Let $|A|$ denote the Lebesgue measure of a measurable subset $A$ of $\\mathbb{R}\^3$. We can thus estimate
> > > > > \\begin{align*}
> > > > >     ||\\sinh(f)||\^2\_{L\^2} &= \\int\_{\\{|f| \\leq 1\\}} |\\sinh(f)|\^2+ \\int\_{\\{|f| > 1\\}} |\\sinh(f)|\^2 \\\\
> > > > >     & \\leq \\cosh(1)\^2 \\int\_{\\{|f| \\leq 1\\}} |f|\^2+ |\\{|f| > 1\\}| (\\sinh( C ||f||\_{H\^2}))\^2 \\\\
> > > > >     & \\leq \\cosh(1)\^2  ||f||\_{H\^2}\^2+ |\\{|f| > 1\\}| (\\sinh( C ||f||\_{H\^2}))\^2
> > > > > \\end{align*}
> > > > > where $|\\{|f| > 1\\}|$ can be bounded by
> > > > > \\[
> > > > >     |\\{|f| > 1\\}| = \\int\\limits\_{\\{|f| > 1\\}} 1 \\leq \\int\\limits\_{\\{|f| > 1\\}} |f|\^2 \\leq ||f||\^2\_{L\^2} \\leq ||f||\^2\_{H\^2}
> > > > > \\]
> > > > > which is finite as $f \\in H\^2(\\mathbb{R}\^3)$.
> > > > >
> > > > > To prove continuity, we again only have to consider the $\\sinh$ term, since the continuity of $-\\Delta$ follows from the definition of $H\^2$. We proceed as above: let $(f\_n)\_n \\subset H\^2(\\mathbb{R}\^3)$ be a sequence converging in $H\^2(\\mathbb{R}\^3)$ and $M = \\sup\_{n \\in \\mathbb{N}} ||f\_n||\_{H\^2}$ (which is finite for otherwise $(f\_n)\_n$ would not be convergent). Let $\\epsilon > 0$ and $N\_1 \\in \\mathbb{N}$ be large enough such that for $n \\geq N\_1$
> > > > > \\[
> > > > >     |\\{ |f - f\_n| > 1 \\}| < \\frac{\\epsilon}{\\sqrt{8} \\sinh(C M)}
> > > > > \\]
> > > > > which is again existing, since $(f\_n)\_n$ converges to $f$ in $H\^2$ and thus $L\^2$. Furthermore, let $N\_2 \\in \\mathbb{N}$, s.t.\\ for $n \\geq N\_2$
> > > > > \\[
> > > > >     || f- f\_n ||\_{H\^2} < \\frac{\\epsilon}{\\sqrt{2} \\cosh(1)} \\; .
> > > > > \\]
> > > > > Noting that $\\sinh$ is Lipschitz continuous with constant $\\cosh(1)$ on $[-1,1]$, it follows that for all $n \\geq \\max\\{N\_1,N\_2\\}$
> > > > > \\begin{align*}
> > > > >     ||\\sinh(f)-\\sinh(f\_n)||\^2\_{L\^2} &= \\int\_{\\{|f-f\_n| \\leq 1\\}} |\\sinh(f)-\\sinh(f\_n)|\^2+ \\int\_{\\{|f-f\_n| > 1\\}} |\\sinh(f)-\\sinh(f\_n)|\^2 \\\\
> > > > >     & \\leq \\cosh(1)\^2 \\int\_{\\{|f-f\_n| \\leq 1\\}} |f-f\_n|\^2+ \\int\_{\\{|f-f\_n| > 1\\}} \\left( |\\sinh(f)|+|\\sinh(f\_n)|\\right)\^2 \\\\
> > > > >     & \\leq \\cosh(1)\^2  ||f-f\_n||\^2\_{H\^2}+ |\\{|f-f\_n| > 1\\}| 4 \\sinh( C M)\^2 < \\\\
> > > > >     & < \\cosh(1)\^2 \\frac{\\epsilon\^2}{2 \\cosh(1)\^2} + \\frac{\\epsilon}{8 (\\sinh(CM))\^2} 4 (\\sinh(CM))\^2 = \\epsilon\^2 \\; .
> > > > > \\end{align*}
> > > > > Analogously we check that the function ${\\bf F}$ is Frechet differentiable with Frechet derivative $\\mathrm D{\\bf F}[f] \\colon H\^2(\\mathbb{R}\^3) \\to L\^2(\\mathbb{R}\^3)$ with
> > > > > \\[
> > > > >    \\mathrm D{\\bf F}[f] = -\\Delta + \\cosh(f) \\; .
> > > > > \\]
> > > > >
> > > > > Since $-\\Delta$ is linear, we only need to check that it holds for the term $\\sinh(f)$. We use Taylor's theorem to rewrite $\\sinh(h)-\\sinh(f)$ pointwise (remember that $H\^2(\\mathbb{R}\^3) \\subseteq \\mathscr C(\\mathbb{R}\^3)$),
> > > > > \\[
> > > > >     \\sinh(h)-\\sinh(f) = \\cosh(f) (h-f)  + (h-f)\^2 \\int\\limits\_0\^1 (1-t) \\sinh(t(h-f)+f) d t.
> > > > > \\]
> > > > > Using the fact that the function $x \\mapsto |\\sinh(x)|$ is convex, we find that for $t \\in [0,1]$
> > > > > \\[
> > > > >     | \\sinh(t(h-f)+f) | \\leq \\max\\{ |\\sinh(h)|, |\\sinh(f)|\\}
> > > > > \\]
> > > > > and thus estimate the integral in the $L\^\\infty$ norm by
> > > > > \\begin{align*}
> > > > >     \\biggl||\\int\_0\^1 (1-t) \\sinh(t(h-f)+f) d t\\biggr||\_{L\^\\infty} &\\leq \\max\\left\\{ ||\\sinh(h)||\_{L\^\\infty}, ||\\sinh(f)||\_{L\^\\infty} \\right\\} \\int\\limits\_0\^1 (1-t) d t \\leq \\\\
> > > > >     &\\leq \\frac{1}{2} \\max\\left\\{ \\sinh\\left(||f||\_{L\^\\infty}\\right), \\sinh\\left(||h||\_{L\^\\infty}\\right) \\right\\} \\\\
> > > > >     &\\leq \\frac{1}{2} \\max\\left\\{ \\sinh\\left(||f||\_{H\^2}\\right), \\sinh\\left(||h||\_{H\^2}\\right) \\right\\} \\; .
> > > > > \\end{align*}
> > > > > Calling this constant $C(f,h)$, we note that it can be bounded uniformly by a constant $D > 0$ in any given ball $B\_R(f)$ of radius $R > 0$ around $f$, since $||h||\_{H\^2} \\leq ||f||\_{H\^2} + R$ for all $h \\in B\_R(f)$. We thus find that
> > > > > \\begin{align*}
> > > > >     ||\\sinh(h)-\\sinh(f)-\\cosh(f)(h-f)||\_{L\^2}\^2 & \\leq \\biggl||\\int\_0\^1 (1-t) \\sinh(t(h-f)+f) d t\\biggr||\_{L\^\\infty} \\int |f-h|\^4 \\leq \\\\
> > > > >     & \\leq C(f,h) ||f-h||\_{L\^4}\^4 \\leq (C')\^4 D ||f-h||\_{H\^2}\^4
> > > > > \\end{align*}
> > > > > where we again applied the Sobolev inequalities yielding a constant $C'$ s.t.\\ \\[
> > > > >     ||h||\_{L\^4} \\leq C' ||h||\_{H\^2},
> > > > > \\]
> > > > > cf.\\ [1,Theorem 4.12]. This proves that ${\\bf F}$ is everywhere Frechet differentiable, as we have shown that
> > > > > \\[
> > > > >     ||\\sinh(h)-\\sinh(f)-\\cosh(f)(h-f)||\_{L\^2} \\leq  (C')\^2 \\sqrt{D} ||f-h||\_{H\^2}\^2 = o \\left(||f-h||\_{H\^2} \\right) \\; .
> > > > > \\]

---

> > > > > > ### Author Response · Authors · 2022-12-02
> > > > > > **Proof continued**
> > > > > >
> > > > > >
> > > > > >
> > > > > > To show that $f \\mapsto \\mathrm{D}{\\bf F}[f]$ is continuous, one only needs to show that $f \\mapsto \\cosh(f)$ is continuous as a map from $H\^2(\\mathbb{R}\^3) \\to L\^\\infty(\\mathbb{R}\^3)$ (where we consider $L\^\\infty(\\mathbb{R}\^3)$ as a subspace of the continuous multiplication operators from $H\^2(\\mathbb{R}\^3) \\to L\^2(\\mathbb{R}\^3)$). This can be done by exploiting the Sobolev inequality since $H\^2$ convergence implies $L\^\\infty$ convergence. Then, for $f,h \\in H\^2$, s.t.\\ $||f-h||\_{L\^\\infty} < 1$, we can exploit the Lipschitz continuity of $\\cosh$ on $[-1,1]$ with constant $\\sinh(1)$  to conclude
> > > > > > \\[
> > > > > >     || \\cosh(f)- \\cosh(h)||\_{L\^\\infty} \\leq \\sinh(1) ||f-h||\_{L\^\\infty} \\leq \\sinh(1) C ||f-h||\_{H\^2} \\;.
> > > > > > \\]
> > > > > >
> > > > > > Finally, $\\mathrm D{\\bf F}[f]$ is invertible with continuous inverse for every $f \\in H\^2(\\mathbb{R}\^3)$. To see this, consider $A := -\\Delta + \\cosh(f)$ as a densely defined unbounded operator on $L\^2$. As $\\cosh(f)$ is a real-valued bounded multiplication operator, because $f$ is a bounded function, it is self-adjoint and thus by the Kato-Rellich Theorem, [2, Theorem X.12], $A$ is self-adjoint with domain $D(A) = D(-\\Delta) = H\^2(\\mathbb{R}\^3)$ and bounded from below.
> > > > > >
> > > > > > From the Min-Max principle, [3, Theorem XIII.1], it follows that
> > > > > > \\[
> > > > > >     \\inf \\text{Spec}(A) = \\inf\_{\\substack{v \\in D(A)\\\\ ||v||\_{L\^2} = 1}} \\Braket{v,Av}
> > > > > > \\]
> > > > > > but
> > > > > > \\[
> > > > > >     \\Braket{v,Av}\_{L\^2} = \\underbrace{\\Braket{\\nabla v, \\nabla v}\_{L\^2}}\_{ \\geq 0} + \\bigl\\langle v, \\underbrace{\\cosh(f)}\_{\\geq 1} v \\bigr\\rangle\_{L\^2} \\geq \\Braket{v,v}\_{L\^2} \\;
> > > > > > \\]
> > > > > > i.e.\\ $\\inf \\text{Spec}(A) \\geq 1$. In particular, $0 \\notin \\text{Spec}(A)$, which means that $A\^{-1}$ is a bounded operator, mapping $L\^2(\\mathbb{R}\^3)$ into $D(A) = H\^2(\\mathbb{R}\^3)$. As a map $L\^2(\\mathbb{R}\^3) \\to L\^2(\\mathbb{R}\^3)$ this map has operator norm $||A\^{-1}||\_{L\^2; L\^2} \\leq 1$ as
> > > > > > \\[
> > > > > >     ||A\^{-1}||\_{L\^2; L\^2} = \\sup \\text{Spec}(A\^{-1}) = \\sup \\left\\{ \\lambda \\in \\mathbb{R} \\, \\big| \\, \\lambda\^{-1} \\in \\text{Spec}(A) \\right\\} \\leq \\sup (0,1] = 1
> > > > > > \\]
> > > > > >
> > > > > > As a map $L\^2(\\mathbb{R}\^3) \\to H\^2(\\mathbb{R}\^3)$, the operator $A\^{-1}$ is bounded as well. Using the following definition of the norm on $H\^2$
> > > > > > \\[
> > > > > >     || v ||\_{H\^2} := || (-\\Delta+1) v||\_{L\^2}
> > > > > > \\]
> > > > > > we have
> > > > > > \\begin{align*}
> > > > > >     || A\^{-1} v||\_{H\^2} & = || (-\\Delta +1 ) A\^{-1} v ||\_{L\^2}
> > > > > >     = || (-\\Delta + \\cosh(f) +1 - \\cosh(f) ) A\^{-1} v ||\_{L\^2} \\\\
> > > > > >     & \\leq || \\underbrace{(-\\Delta + \\cosh(f)) A\^{-1}}\_{ = 1} v ||\_{L\^2} +|| (1 - \\cosh(f) ) A\^{-1} v||\_{L\^2} \\\\
> > > > > >     & \\leq ||v||\_{L\^2} + || 1-\\cosh(f) ||\_{L\^\\infty} \\underbrace{||A\^{-1} v||}\_{\\leq ||v||} \\leq \\left( 1  + || 1-\\cosh(f) ||\_{L\^\\infty} \\right) ||v||
> > > > > > \\end{align*}
> > > > > > where we used that $1-\\cosh(f)$ is a bounded operator as $f \\in H\^2(\\mathbb{R}\^3) \\subset L\^\\infty(\\mathbb{R}\^3)$. This shows that $A\^{-1}$ is a bounded operator with norm $||A\^{-1}||\_{L\^2; H\^2} \\leq 1  + || 1-\\cosh(f) ||\_{L\^\\infty} $.
> > > > > >
> > > > > > [1] Robert A. Adams and John J. F. Fournier. Sobolev spaces, volume 140 of Pure and Applied Mathematics (Amsterdam). Elsevier/Academic Press, Amsterdam, second edition, 2003
> > > > > >
> > > > > > [2] Michael Reed and Barry Simon. Methods of modern mathematical physics. II. Fourier analysis,
> > > > > > self-adjointness. Academic Press [Harcourt Brace Jovanovich, Publishers], New York-London,
> > > > > > 1975.
> > > > > >
> > > > > > [3] Michael Reed and Barry Simon. Methods of modern mathematical physics. IV. Analysis of
> > > > > > operators. Academic Press [Harcourt Brace Jovanovich, Publishers], New York-London, 1978

---

> > > > > > > ### Comment · Reviewer_cgGX · 2022-12-03
> > > > > > > **Thank you for your response**
> > > > > > >
> > > > > > > Thank you for your response. To be very honest, I have a hard time seeing how a "slight typo" can change "H^2" to "W^{2, \infty}" and, simultaneously, "L^{2}" to "W^{0, \infty}." The new regularity result is, as far as I can tell, correct, although it does not yet cover the case of Dirichlet boundary conditions presented in the experimental section.
> > > > > > >
> > > > > > > I am also still not convinced by
> > > > > > >
> > > > > > > >Theorem 4 explicitly proves why the shift in ambiguity is not a problem for systems where original assumptions hold. For systems they don't hold in, the issue is only as bad as the one in the usual method: as such, any improvement error correction can provide necessarily leads to a better outcome.
> > > > > > >
> > > > > > > After all, there are essentially two scenarios: Either the inverse/implicit function theorem holds, and therefore the original loss is a good proxy for accuracy. In this case, the uncorrected method already provides an "error" estimate. Alternatively, the above is not true, in which case there is nothing that can be said about the higher-order corrected method. Thus, I see no theoretical justification for why the error estimates of the correction should be more accurate in describing overall error than the original error estimate. Besides, where is $$\sigma_{\min_{i}}$$ defined?
> > > > > > >
> > > > > > > Regarding the ablation experiments, they do not seem to contain the case of a sum of two neural networks (basically the same structure as used by the correction framework without the two-scale training procedure)
> > > > > > >
> > > > > > > Overall, I congratulate the authors to their exciting empirical results, but I believe that the paper will require substantial rewriting before being suitable for publication in ICLR or similar venues. I, therefore, continue to recommend the rejection of the paper.

---

> > > > > > > > ### Author Response · Authors · 2022-12-05
> > > > > > > > **Response to Reviewer cgGX**
> > > > > > > >
> > > > > > > > We thank you again for outlining your concerns as you see them. We have aimed a full response here, excepting the point related to the inverse function theorem. That will require a more detailed/tangential response like the regularity one, which we had omitted for the same reasons and we would be grateful if you could consider it in a few days' time.
> > > > > > > >
> > > > > > > > 1: "To be very honest, I have a hard time seeing how a "slight typo" can change "$H\^2$" to "$W\^{2, \\infty}$" and, simultaneously, "$L^{2}$" to "$W^{0, \infty}$"
> > > > > > > >
> > > > > > > > The originally quoted map was from $W^{2, \infty} \mapsto W^{0, \infty}$. The current/corrected map is from $W^{2, 2} \mapsto W^{0, 2}$ ($H^2$ and $L^2$ are simply shorter names for those spaces, hence the switch in notation). We simply had an $\infty$ floating around from prior versions instead of a $2$: the fact that the change has no bearing on any result or discussion of the paper at all, except correcting the error itself, is why it is a minor adjustment.
> > > > > > > >
> > > > > > > >
> > > > > > > > 2: "The new regularity result is, as far as I can tell, correct, although it does not yet cover the case of Dirichlet boundary conditions presented in the experimental section."
> > > > > > > >
> > > > > > > > Let $\mathcal{D}$ be any compact domain with a locally Lipschitz boundary $\partial \mathcal{D}$ under the assumptions for which the result was quoted. For any such $\mathcal{D}$, our proof holds verbatim, if one replaces $H^2(\mathbb{R}^3)$ and $L^2(\mathbb{R}^3)$ with $H^2_0(\mathcal{D})$ (the Dirichlet boundary condition subspace of $H^2(\mathcal{D})$) and $L^2(\mathcal{D})$, as the same Sobolev inequalities hold for these spaces, cf. Part III of [1, Theorem 4.12]. Vis a vis the numerical section, the same holds for the presented 2D problem as well. We believe it should hold for the 4D version as well, but since the source of the problem is a 3D setting, we did not explore for proofs beyond 3D (and only claimed it for 3D).
> > > > > > > >
> > > > > > > > 3. "where is $\sigma_{min_i}$ defined?"
> > > > > > > >
> > > > > > > > It is defined in the same manner as $\sigma_{min}$, with $\mathrm{D} {\\bf F}\_i$ replacing $\mathrm{D} {\bf F}$ and $\Phi_{\epsilon_i}$ replacing $\Phi$. We will add it to the text explicitly.
> > > > > > > >
> > > > > > > > 4. "Regarding the ablation ---- without the two-scale training procedure)"
> > > > > > > >
> > > > > > > > We apologize for mistaking the asked for experiment: we constructed experiments to answer whether $larger$ models would provide sufficient improvements (which we believe was the test asked for by other reviewers), believing it to be an equivalent test. We will report results on these experiments as soon as possible.
> > > > > > > >
> > > > > > > >
> > > > > > > > 5. "After all, there are essentially two scenarios: Either the inverse/implicit function theorem holds --- the error estimates of the correction should be more accurate in describing overall error than the original error estimate"
> > > > > > > >
> > > > > > > > We shall be answering this point in a few days, since we need to formalize the argument the same way we had to for the regularity one. It was omitted for similar reasons relating to relevance: such arguments will be happily added to the manuscript if the reviewers judge that to be a wise decision.
> > > > > > > >
> > > > > > > > We end by thanking you for recognizing the value of our empirical results.

---

> > > > > > > > > ### Comment · Reviewer_cgGX · 2022-12-05
> > > > > > > > > **Check your claims**
> > > > > > > > >
> > > > > > > > > 2. Is false, your "norm" for H^2 is equivalent to the actual H^2 norm in this case.

---

> > > > > > > > > > ### Author Response · Authors · 2022-12-05
> > > > > > > > > > **Checked claims**
> > > > > > > > > >
> > > > > > > > > > It is immaterial which one of a set of equivalent norms one chooses, as they all induce the same topology. The explicit constants of the Sobolev inequalities depend on the choice of norm, but since we made no assertions about the constants, this is completely irrelevant to our argument.
> > > > > > > > > >
> > > > > > > > > > We would be grateful if you could clarify your question in case our response is not what you meant.

---

> > > > > > > > > > > ### Comment · Reviewer_cgGX · 2022-12-05
> > > > > > > > > > > **Sorry I meant is not equivalent**
> > > > > > > > > > >
> > > > > > > > > > > Sorry I meant is not equivalent

---

> > > > > > > > > > > > ### Author Response · Authors · 2022-12-06
> > > > > > > > > > > > **Proof of Equivalence of Norms**
> > > > > > > > > > > >
> > > > > > > > > > > > Unless we have mistaken the point of your question, the norms should be equivalent. The proof is below. Please let us know in case this is not what you are referring to:
> > > > > > > > > > > >
> > > > > > > > > > > > Let $\\Omega \\subset \\mathbb{R}\^3$ be an open domain with compact closure. $H\^2\_0(\\Omega)$ is the closure of $\\mathcal{C}\^\\infty\_c(\\Omega)$, the compactly supported smooth function on $\\Omega$ w.r.t. the norm
> > > > > > > > > > > > \\[
> > > > > > > > > > > >     \\| f \\|\_{H\^2} := \\Big( \\|f\\|\^2\_{L\^2} + \\sum\_{i = 1}\^3 \\| \\partial\_i f\\|\_{L\^2}\^2 + \\sum\_{i,j = 1}\^3 \\|\\partial\_i, \\partial\_j f\\|\_{L\^2}\^2 \\Big)\^{\\frac{1}{2}}.
> > > > > > > > > > > > \\]
> > > > > > > > > > > >
> > > > > > > > > > > > Furthermore, let
> > > > > > > > > > > > \\[
> > > > > > > > > > > >     \\| f \\|\_* := \\|(-\\Delta+1)f\\|\_{L\^2} .
> > > > > > > > > > > > \\]
> > > > > > > > > > > >
> > > > > > > > > > > > The two norms $\\| \\cdot \\|\_{H\^2}$ and $\\| \\cdot \\|\_*$ are equivalent on $H\^2\_0(\\Omega)$. Since by definition $\\mathcal{C}\^\\infty\_c(\\Omega)$ is dense in $H\^2\_0(\\Omega)$, it is enough to show this for all $f \\in \\mathcal{C}\^\\infty\_c(\\Omega)$. We can extend the domain of integration in the $L\^2$ norms w.l.o.g. to all of $\\mathbb{R}\^3$ since $f$ and all of its derivatives are compactly supported. Thus, we can perform integration by parts, since $f$ and its derivatives are smooth and compactly supported, which means for functions $f,g$ appearing below one has
> > > > > > > > > > > > \\[
> > > > > > > > > > > >     \\Braket{f, \\partial\_i g}\_{L\^2} =  \\Braket{- \\partial\_i f, g}\_{L\^2}.
> > > > > > > > > > > > \\]
> > > > > > > > > > > >
> > > > > > > > > > > > Using this we find
> > > > > > > > > > > >
> > > > > > > > > > > > \\[\\|f\\|\_*\^2 = \\Braket{(1-\\Delta)f,(1-\\Delta)f}\_{L\^2} = \\sum\_{i,j = 1}\^3 \\Braket{(1-\\partial\_i \\partial\_i)f,(1-\\partial\_j\\partial\_j)f}\_{L\^2}\\]
> > > > > > > > > > > >
> > > > > > > > > > > > \\[= \\sum\_{i,j = 1}\^3 \\left( \\Braket{f,f}\_{L\^2}+ \\Braket{-\\partial\_i \\partial\_i f,f}\_{L\^2} + \\Braket{f,-\\partial\_j\\partial\_j f}\_{L\^2} + \\Braket{\\partial\_i \\partial\_i f,\\partial\_j\\partial\_j f}\_{L\^2} \\right)\\]
> > > > > > > > > > > >
> > > > > > > > > > > > \\[= \\sum\_{i,j = 1}\^3 \\left( \\Braket{f,f}\_{L\^2}+ \\Braket{\\partial\_i  f, \\partial\_i f}\_{L\^2} + \\Braket{\\partial\_j f,\\partial\_j f}\_{L\^2} + \\Braket{\\partial\_j \\partial\_i f,\\partial\_j\\partial\_i f}\_{L\^2} \\right)\\]
> > > > > > > > > > > >
> > > > > > > > > > > > \\[= 9 \\|f\\|\_{L\^2}\^2 + 4 \\sum\_{i = 1}\^3 \\| \\partial\_i f\\|\_{L\^2}\^2 + \\sum\_{i,j = 1}\^3 \\| \\partial\_i \\partial\_j f\\|\_{L\^2}\^2
> > > > > > > > > > > > \\]
> > > > > > > > > > > >
> > > > > > > > > > > > It therefore follows that $\\|f\\|\_{H\^2} \\leq \\|f\\|\_* \\leq  3 \\|f\\|\_{H\^2}$, i.e. the two norms are equivalent.

---

> > > > > > > > > > > > > ### Comment · Reviewer_cgGX · 2022-12-06
> > > > > > > > > > > > > **Mea culpa**
> > > > > > > > > > > > >
> > > > > > > > > > > > > My bad, I was thinking about $H^2 \cap H_0^1$ instead of $H^2_0$. The problem with $H^2_0 \rightarrow L^2$ is that $L^2$ functions can be arbitrarily large close to the boundary, whereas the second derivatives of $H^2_{0}$ (and thus, their Laplace operators) have to vanish as the boundary is approached. Thus, once again, the map can not be invertible between these spaces. In general, the so-called "higher regularity" you have in mind requires additional regularity assumptions on the boundary beyond Lipschitz continuity. I recommend you look these things up. Unfortunately, I will likely not have time to discuss these matters further.

---

> > > > > > > > > > > > > > ### Author Response · Authors · 2022-12-07
> > > > > > > > > > > > > > **Sine Cura Sis and responding to the invertibility point**
> > > > > > > > > > > > > >
> > > > > > > > > > > > > > Thanks for taking the time to go deep into our work. We understand if you can't respond to these points anymore, but we are certain the argument is robust (we took your advice to look things up and found nothing that contradicts our belief). Could you perhaps let us know explicitly where our reasoning is explicitly wrong?
> > > > > > > > > > > > > >
> > > > > > > > > > > > > > What you are saying seems to contradict well-known mathematical results on the Sobolev inequalities for domains with locally Lipschitz boundary, the Kato-Rellich theorem or the invertibility of (strictly!) positive operators, for all of which we have given references.
> > > > > > > > > > > > > >
> > > > > > > > > > > > > > Further, since $H^2_0(\Omega)$ is a subset of only $C^{1/2}(\Omega)$, there is absolutely no reason for the second derivatives to have a well-defined trace to the boundary or to vanish there, e.g. the function $f(x,y,z) = 1-\max\\{x^2,y^2,z^2\\}$ is in $H^2_0( (-1,1)^3 )$ but has constant Laplacian -2.

---

> > > > > > > > > > > > > > > ### Comment · Reviewer_cgGX · 2022-12-07
> > > > > > > > > > > > > > > **Sorry I was imprecise**
> > > > > > > > > > > > > > >
> > > > > > > > > > > > > > > Sorry, my reasoning was a little flawed but I don't think the conclusion is:
> > > > > > > > > > > > > > >
> > > > > > > > > > > > > > > consider for simplicity the function $u(x)$ defined by $x \mapsto 1 - x^3$ on $(-1,1)$. By the Sobolev embedding theorem, this means that the first derivative of $u$ is hoelder continuous (I believe the exponent is 1/2). Now, in particular, the H^2 norm upper bounds the corresponding holder norm. Now try to approximate the function $u$ with compactly supported functions: The problem is that the holder norm of first derivative of the mismatch is lower bounded away from $0$, and thus the $H^2$ norm is, making it impossible to approximate $u$ in $H^2$ with compactly supported functions, thus $u$ is not in $H^2_0$

---

> > > > > > > > > > > > > > > > ### Author Response · Authors · 2022-12-08
> > > > > > > > > > > > > > > > **Clarification**
> > > > > > > > > > > > > > > >
> > > > > > > > > > > > > > > > Thanks for taking the time and no need to apologize.
> > > > > > > > > > > > > > > >
> > > > > > > > > > > > > > > > We admit we are confused about your latest points: we acknowledge that your $u$ is not in $Hˆ2\_0$, but don't see the point being made or how our result could be flawed. To clarify what we meant about how constraint condition case is handled, we have devised this alternate argument that might possibly be more clarifying.
> > > > > > > > > > > > > > > >
> > > > > > > > > > > > > > > > We remark on a way to add linear constraints (of which 0 boundary conditions are an example) on the elements of $G$ that will still allow us to apply the Theorems directly. For this purpose let $E$ be an auxiliary Hilbert space and the constraint map $\\mathfrak{C}\\colon G \\to E$ be a bounded linear map encoding a given constraint by requiring that our solution is in $\\mathfrak{C}\^{-1}(0)$ which is a closed subspace of $E$. Examples of such maps include restrictions to subsets of our domain of interest or imposing boundary conditions if the trace map onto the boundary its continuous. Let $U\_{\\mathfrak{C}} := U \\cap \\mathfrak{C}\^{-1}(0)$, $P : G \\to  \\mathfrak{C}\^{-1}(0)$ be the orthogonal projection onto the closed subspace $ \\mathfrak{C}\^{-1}(0)$ and the $P\^\\dagger :  \\mathfrak{C}\^{-1}(0) \\to G$ the canonical inclusion map. The new loss function then is given by $\\mathcal{L}\_{\\mathfrak{C}} : U\_{\\mathfrak{C}} \\to \\mathbb{R}$
> > > > > > > > > > > > > > > >
> > > > > > > > > > > > > > > > \\[
> > > > > > > > > > > > > > > >     \\mathcal{L}\_{\\mathfrak{C}}(\\mathcal{N}) = \\Braket{{\\bf F}(P\^\\dagger \\mathcal{N}), {\\bf F}(P\^\\dagger \\mathcal{N})}
> > > > > > > > > > > > > > > > \\]
> > > > > > > > > > > > > > > >
> > > > > > > > > > > > > > > > which yields the gradient descent equation
> > > > > > > > > > > > > > > >
> > > > > > > > > > > > > > > > \\[
> > > > > > > > > > > > > > > >     \\dot{\\mathcal{N}} = - P \\nabla \\mathcal{L}\_{\\mathfrak{C}}(\\mathcal{N}) .
> > > > > > > > > > > > > > > > \\]
> > > > > > > > > > > > > > > >
> > > > > > > > > > > > > > > > $\\sigma\_{\\min,\\mathfrak{C}}$, the convergence rate of the restricted model, satisfies
> > > > > > > > > > > > > > > >
> > > > > > > > > > > > > > > > \\[
> > > > > > > > > > > > > > > >     \\sigma\_{\\min,\\mathfrak{C}} \\geqslant \\sigma\_{\\min}
> > > > > > > > > > > > > > > > \\]
> > > > > > > > > > > > > > > >
> > > > > > > > > > > > > > > > since one is considering the infimum of the spectrum of $\\mathrm{D} {\\bf F}[\\Phi]\^\\dagger\\mathrm{D} {\\bf F}[\\Phi]$ only over the restricted subspace $\\mathfrak{C}\^{-1}(0)$ instead of all of $G$.
> > > > > > > > > > > > > > > >
> > > > > > > > > > > > > > > >
> > > > > > > > > > > > > > > > For the particular case of solving the nPBE on an open domain with compact closure and locally Lipschitz boundary $\\partial \\Omega$, the map $\\mathfrak{C}$ maps $H\^2(\\mathbb{R}\^n)$ into $L\^2(\\mathbb{R}\^n \\setminus \\overline{\\Omega}) \\oplus L\^2(\\partial \\Omega)$. The map into $L\^2(\\mathbb{R}\^n \\setminus \\overline{\\Omega})$ is given by restricting a given function in $\\mathbb{R}\^n$ to $\\mathbb{R}\^n \\setminus  \\overline{\\Omega}$ which is trivially continuous, as one only integrates over a subset of the original domain when computing the $L\^2$ norm of the image and does not take into account the norm of the derivatives.
> > > > > > > > > > > > > > > >
> > > > > > > > > > > > > > > > The map into $L\^2(\\partial \\Omega)$ is the composition of restricting the function to $\\Omega$ and then performing the trace operation onto the boundary $\\partial \\Omega$, which is continuous since the boundary is locally Lipschitz, cf. [Theorem 3.38, McLean] for the stronger result that it is continuous $H\^1(\\Omega) \\to H\^{1/2}(\\partial\\Omega)$, whence our assertion follows since $\\| \\cdot \\|\_{L\^2(\\partial \\Omega)} \\leq \\| \\cdot \\|\_{H\^{1/2}(\\partial \\Omega)}$ and $\\|\\cdot\\|\_{H\^1(\\Omega)} \\leq \\|\\cdot \\|\_{H\^2(\\Omega)} \\leq \\| \\cdot \\|\_{H\^2(\\mathbb{R}\^n)}$.
> > > > > > > > > > > > > > > >
> > > > > > > > > > > > > > > > Requiring that $\\mathfrak{C}(f) = 0$ gives us a version of Dirichlet boundary conditions for the domain $\\Omega$, which allows us to directly apply our theoretical framework, since we have established the result for $H\^2(\\mathbb{R}\^3)$.
> > > > > > > > > > > > > > > >
> > > > > > > > > > > > > > > > We hope this construction alleviates your concern.
> > > > > > > > > > > > > > > >
> > > > > > > > > > > > > > > > [1] William McLean, Strongly elliptic systems and boundary integral equations, Cambridge University Press, Cambridge, 2000

---

> > > > > > > > ### Author Response · Authors · 2022-12-12
> > > > > > > > **Utility of error correction vis a vis systems satisfying or not satisfying Inverse Function Theorem**
> > > > > > > >
> > > > > > > > We thank the reviewer for devoting so much of their time to these discussions. We address the final point left completely unaddressed by us:
> > > > > > > >
> > > > > > > > "Either the inverse/implicit function theorem holds, ----- the error estimates of the correction should be more accurate"
> > > > > > > >
> > > > > > > > We partially agree with the raised sentiments, since that is what Theorems 1 and 2 are all about, and the literature seemed to be missing them. However, the uncorrected method does not $already$ $provide$ $an$ $error$ $estimate$. It gives a weak understanding of what the $upper bound$ on the norm of the error could roughly be. Error correction provides an actual reliable model for the error: one can't simply read off ${\bf F}[\mathcal{N}]$ and convert them into an error model without it. Error analysis and correction for inverse function amenable systems is not a repackaged statement of the current situation - it is a method that adds significantly more information in a reliable/quantifiable manner, and leads to significant performance boosts.
> > > > > > > >
> > > > > > > > Further, assume $\mathrm{D}{\bf F}[\Phi]$ fails to be invertible, but still has an empty kernel (as is the case with differential operators on a space of functions with unbounded domain). Then, one can still apply gradient descent in any finite dimensional approximation (as in, a model $\mathcal{N}$ with a finite number of parameters), as $\sigma_{\min}(\mathcal{N}) = \inf \text{Spec}\Big[(\mathrm{D}{\bf F}[\mathcal{N}])^{\dagger} \mathrm{D}{\bf F}[\mathcal{N}] \Big]$ for that approximation will be strictly positive. However, increasing the number of dimensions will let $\sigma_{\min}(\mathcal{N}) \to 0$, hence leading to a worse speed of convergence when one requires higher accuracy. In these cases, we have to balance the time willing to be spent on training with the required accuracy, as any finite dimensional approximation generically will be able to approach the true solution only up to a fixed distance $\delta$, that does not decrease with the amount of training, only the number of available parameters themselves. Each extra model makes $\delta$ smaller, but also decreases the convergence speed to it. As such, error correction locks in the performance of the standard method, and is capable of doing better than it, but at increasingly worse costs.

---

### Author Response · Authors · 2022-11-19
**General response to all reviewers**

We thank the reviewers for the effort they put into going through our work. The raised points have led us to significantly better presentation and discussions. We are glad to report that the raised criticisms and requests for more information can be handled within this work's framework with minimal adjustments: the manuscript has been updated correspondingly to correct for such shortcomings in the original draft. An additional theorem, guaranteeing the existence of an exponentially convergent phase of optimization, has also been added to clarify the utility and reliability of the proposed methods. Together with the inequalities and the proposed algorithms, we believe our updated work represents a self-contained blueprint for building and analysing NN DE solvers for systems that satisfy the assumptions on ${\bf F}$. We hope the reviewers agree with our conclusions.

The noted weaknesses lay in two major categories:

1. Limited Numerical experiments
2. Flaws in contextualizing and discussing why/when the proposed theorems and algorithms should be expected to work

We summarize the steps taken to address those concerns below (additional reviewer specific comments have been made directly to each reviewer as well):

1: The existing experiments cover a variety of DEs and solutions (including chaotic DEs, nonlinear PDEs, etc), with relevance in various scientific fields (astrophysics, electrochemistry) and are adapted from different codebases/papers (PINNs, HNNs). Additionally, ablation studies are now in place for both size and optimization time. The current set of results promisingly verifies the entire framework we have built for describing NN DE solvers.

However, to eliminate the reasonable doubts raised by multiple reviewers, we continue to work towards incorporating the numerical experiments recently advised by "reviewer 3Hkr" and will report them during phase 2 of discussion, as per their advice. We apologize for not finishing these new examples within the rebuttal submission deadlines, and thank "reviewer 3Hkr" for understanding the delay.

2: All theorems now apply to a larger class of loss functions than before. Additionally, we now have a new theorem that better contexualizes the value of the others: we prove the existence of an exponentially convergent optimization phase for each system satisfying our assumptions. Thus, we obtain a better match between empirical observations about NN DE solvers and the theoretical machinery we have built for to describe them. The discussion around what these results imply for an error correction scheme has been updated to better reflect the larger body of results. We also explicitly give examples of where each result is relevant.

---

### Decision · Program_Chairs · 2023-01-20

**Decision:**

Reject

**Justification For Why Not Higher Score:**

see above.

**Justification For Why Not Lower Score:**

N/A

**Metareview: Summary, Strengths And Weaknesses:**

While some of the reviewers are vaguely positive about this paper (at low confidence), reviewer cgGX (who is an expert) has raised a serious concern about the validity of a central proof, which has not been addressed by the reviewers, despite repeated explicit and clear requests by the reviewer. The paper can thus not be accepted in this form.